# ORIGEN: Zero-Shot 3D Orientation Grounding in Text-to-Image Generation

**Yunhong Min**[*]    **Daehyeon Choi**[*]    **Kyeongmin Yeo**    **Jihyun Lee**    **Minhyuk Sung**

KAIST

{dbsghd363,daehyeonchoi,aaaaa,jyun.lee,mhsung}@kaist.ac.kr

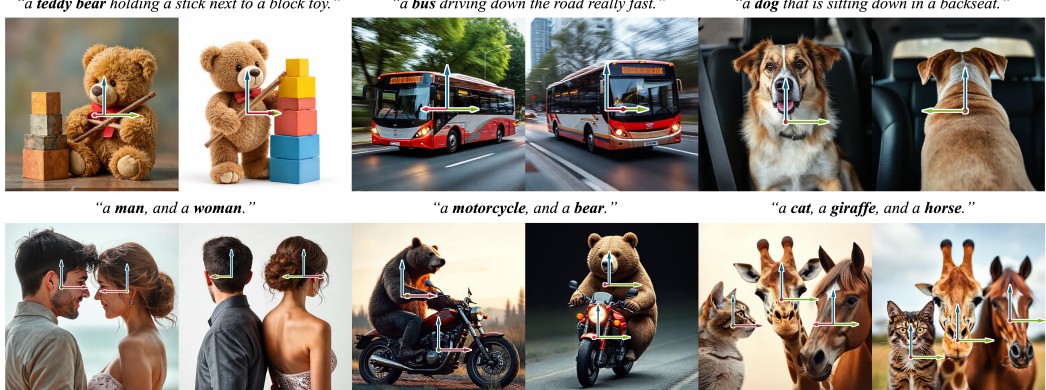

Figure 1: **3D orientation-grounded text-to-image generation results of ORIGEN.** We present ORIGEN, the first zero-shot method for 3D orientation grounding in text-to-image generation across multiple objects and diverse categories. ORIGEN generates high-quality images that are accurately aligned with the grounding orientation conditions (colored arrows) and the input text prompts.

## Abstract

We introduce ORIGEN, the first zero-shot method for 3D orientation grounding in text-to-image generation across multiple objects and diverse categories. While previous work on spatial grounding in image generation has mainly focused on 2D positioning, it lacks control over 3D orientation. To address this, we propose a reward-guided sampling approach using a pretrained discriminative model for 3D orientation estimation and a one-step text-to-image generative flow model. While gradient-ascent-based optimization is a natural choice for reward-based guidance, it struggles to maintain image realism. Instead, we adopt a sampling-based approach using Langevin dynamics, which extends gradient ascent by simply injecting random noise—requiring just a single additional line of code. Additionally, we introduce adaptive time rescaling based on the reward function to accelerate convergence. Our experiments show that ORIGEN outperforms both training-based and test-time guidance methods across quantitative metrics and user studies. Project Page: https://origen2025.github.io.

## 1   Introduction

Controllability is a key aspect of generative models, enabling precise, user-driven outputs. In image generation, *spatial grounding* ensures structured and semantically meaningful results by incorporating conditions that cannot be fully specified through text alone. Recent research integrating spatial instructions, such as bounding boxes [1–5] and segmentation masks [3, 6, 7, 4, 8, 9], has

---

[*]Equal contribution.

39th Conference on Neural Information Processing Systems (NeurIPS 2025).

shown promising results. While these works have advanced 2D spatial control, particularly *positional* constraints, 3D spatial grounding remains largely unexplored. In particular, orientation is essential for defining an object's spatial pose [10–16], yet its integration into conditioning remains an open challenge.

A few existing methods, such as Zero-1-to-3 [17] and Continuous 3D Words [18], support orientation-conditioned image generation. However, Zero-1-to-3 enables only relative orientation control with respect to a reference foreground image, while Continuous 3D Words is limited to single-object images and supports only half-front azimuth control. Moreover, all these models lack realism because they are trained on synthetic data, i.e., multi-view renderings of centered 3D objects, as real-world training images with accurate per-object orientation annotations are not publicly available. In addition, OrientDream [19] supports orientation control via text prompts, but it is restricted to four primitive azimuths (front, left, back, right) and is also limited to single-object images.

To overcome these limitations, we propose ORIGEN, the first method for generalizable 3D orientation grounding in real-world images across multiple objects and diverse categories. We introduce a *zero-shot* approach that leverages test-time guidance from OrientAnything [20], a foundational discriminative model for 3D orientation estimation. Specifically, using a pretrained one-step text-to-image generative model [21] that maps a latent vector to a real image, along with a reward function defined by the discriminative model, our goal is to find a latent vector whose corresponding real image yields a high reward.

A natural approach for this search is gradient-ascent-based optimization [22], but it struggles to keep the latent distribution aligned with the prior (a standard Gaussian), leading to a loss of realism in the generated images. To address this, we introduce a sampling-based approach that balances reward maximization with adherence to the prior latent distribution. Specifically, we propose a novel method that simulates *Langevin dynamics*, where the drift term is determined by our orientation-grounding reward. We further show that its Euler–Maruyama discretization simplifies to a surprisingly simple formulation–an extension of standard gradient ascent with random noise injection, which can be implemented in a single line of code. To further enhance efficiency, we introduce a novel time-rescaling method that adjusts timesteps based on the current reward value, accelerating convergence.

Since no existing method has quantitatively evaluated 3D orientation grounding in text-to-image generation (except for user studies by [18]), we curate a benchmark based on the MS-COCO dataset [23], mixing and matching object classes and orientations to create images with single or multiple orientation-grounded objects. We demonstrate that ORIGEN significantly outperforms previous orientation-conditioned image generative models [18, 17] on both our benchmark and user studies. Since prior models cannot condition on multiple objects (whereas ORIGEN can, as shown in Fig. 1), comparisons are conducted under single-object conditioning. Additionally, we perform experiments to further validate the superior performance of our method over text-to-image generative models with orientation-specific prompts and other training-free guided sampling strategies.

Overall, our main contributions are:

- We present ORIGEN, the first method for 3D orientation grounding in text-to-image generation for multiple objects across diverse categories.
- We introduce a novel reward-guided sampling approach based on *Langevin dynamics* that provides a theoretical guarantee for convergence while simply adding a single line of code.
- We also propose a reward-adaptive time rescaling method that accelerates convergence.
- We show that ORIGEN achieves significantly better 3D orientation grounding than existing orientation-conditioned image generative models [18, 17], text-to-image generative models [24, 25, 21] with orientation-specific prompts, and training-free guided sampling strategies.

## 2   Related Work

**Viewpoint or Orientation Control.**    Several works have focused on controlling the *global viewpoint* of the entire image. For example, Burgess *et al.* [26] propose a view-mapping network that predicts a word embedding to control the viewpoint in text-to-image generation. Kumari *et al.* [27] enable model customization to modify object properties via text prompts, with added viewpoint control. However, these methods cannot individually control the orientation of foreground objects. Other works have attempted to control *single-object orientation* in image generation. For instance, Liu *et*

*al.* [17] introduce an image diffusion model that controls the *relative* orientation of an object with respect to its reference image. Huang *et al.* [19] propose an orientation-conditioned image diffusion model for sampling multi-view object images for text-to-3D generation. The most recent work in this domain, Cheng *et al.* [18], aim to control object attributes, including azimuth, through continuous word embeddings. However, these methods rely on training-based approaches using single-object synthetic training images, limiting their generalizability across multiple objects and diverse categories. We additionally note that a few works address image generation conditioned on *depth* [28–30] or *3D bounding boxes* [31], but they do not allow direct control of object orientations — for example, 3D bounding boxes have front-back ambiguities.

**Training-Free Guided Generation.** A number of *training-free* methods have been proposed for guided generation tasks. DPS [32], MPGD [33], and Pi-GDM [34] update the noisy data point at each step using a given reward function. FreeDoM [35] takes this further by introducing *rewinding*, where intermediate data points are regenerated by reversing the generative denoising process. The core principle of this approach is leveraging the posterior mean from a noisy image at an intermediate step via Tweedie's formula [36]. The expected future reward can also be computed from the posterior mean, allowing gradient ascent to update the noisy image. However, a key limitation of these methods is that the posterior mean from a diffusion model is often too blurry to accurately predict future rewards. While distilled or fine-tuned diffusion [37–40] and flow models [41, 42] can mitigate this issue, their straightened trajectories lead to insufficiently small updates to the image during gradient ascent at intermediate timesteps.

Recent approaches [22, 43–45] attempt to address this limitation by updating the *initial* noise rather than the intermediate noise. Notably, ReNO [22] is an optimization method based on a *one-step* generative model to efficiently iterate the initial noise update through one-step generation and future reward computation. However, gradient ascent with respect to the initial noise often suffers from local optima and leads to deviations from real images, even with heuristic regularization [46]. To overcome this limitation, we propose a novel sampling-based approach rather than an optimization-based one, leveraging *Langevin dynamics* to effectively balance reward maximization and realism.

**Training-Based Guided Generation.** For controlling image generation, ControlNet [28] and IP-Adapter [47] are commonly used to utilize a pretrained generative model to control for various conditions, though they require training data. For 3D orientation grounding, no public real-world training data is available, and collecting such data would be particularly challenging, especially for multi-object grounding, due to the need for diversity. To address this, recent works have introduced RL-based reward fine-tuning approaches [48? –51]. However, these methods require substantial computational resources and, more importantly, cause significant deviation from the pretrained data distribution when trained with task-specific rewards, leading to degraded image quality and reduced diversity [52]. Hence, instead of fine-tuning, we propose a test-time reward-guided framework that leverages a discriminative foundational model for guidance.

## 3 ORIGEN

We present ORIGEN, a *zero-shot* method for 3D orientation grounding in text-to-image generation. To the best of our knowledge, this is the first 3D orientation grounding method for *multiple* objects across open-vocabulary categories.

### 3.1 Problem Definition and Overview

Our goal is to achieve 3D orientation grounding in text-to-image generation using a *one-step* generative flow model [53] $\mathcal{F}_\theta$, based on *test-time* guidance without fine-tuning the model. Let $\mathbf{I} = \mathcal{F}_\theta(\mathbf{x}, c)$ denote an image generated by $\mathcal{F}_\theta$ given an input text $c$ and a latent $\mathbf{x}$ sampled from a prior distribution $q = \mathcal{N}(\mathbf{0}, \mathbf{I})$. The input text $c$ prompts the generation of an image containing a set of desired objects (e.g., "*A person* in a brown suit is directing *a dog*"). For each of the $N$ objects that appear in $c$, we associate a set of object phrases $\mathcal{W} = \{w_i\}_{i=1}^N$ (e.g., {"*dog*", "*person*"}) and a corresponding set of 3D orientation grounding conditions $\Phi = \{\phi_i\}_{i=1}^N$. Following the convention [20], each object orientation $\phi_i$ is represented by three Euler angles $\phi_i = \{\phi_{i,j}\}_{j=1}^3$: azimuth $\phi_{i,1} \in [0, 360)$, polar angle $\phi_{i,2} \in [0, 180)$, and rotation angle $\phi_{i,3} \in [0, 360)$. The goal of 3D orientation grounding is to generate an image $\mathbf{I}$ in which the $N$ objects appear with their corresponding orientations $\Phi = \{\phi_i\}_{i=1}^N$.

**Why Test-Time Guidance?** Supervised methods are not applicable to this task due to the lack of training datasets containing diverse real-world images with per-object orientation annotations. Previous supervised approaches [18, 17] have therefore been limited to single-object scenarios within a narrow set of categories. To address this, we propose a training-free approach that leverages recent foundational models to design a reward function: GroundingDINO [54] for object detection and OrientAnything [20] for orientation estimation. While this reward function could also be used in recent RL-based self-supervised methods for fine-tuning the image generative model, such techniques not only demand substantial computational resources but also lead to degraded image quality when trained with such a task-specific reward [52]. To this end, we propose a novel training-free approach that effectively avoids image quality degradation while maximizing orientation reward at test time.

In particular, given a reward function $\mathcal{R}$ defined based on GroundingDINO and OrientAnything (which we will discuss in detail in Sec. 3.2), our goal is to find a latent sample $\mathbf{x}$ that maximizes the reward of its corresponding image, expressed as $\mathcal{R}(\mathcal{F}_\theta(\mathbf{x}, c))$. For simplicity, we define the pullback of the reward function as $\hat{\mathcal{R}} = \mathcal{R} \circ \mathcal{F}_\theta$ and use this notation throughout.

**Why Based on a One-Step Generative Model?** Despite extensive research on test-time reward-guided generation using pretrained diffusion models [32, 33, 35], we find that the main challenge stems from the multi-step nature of these models. Reward-guided generation requires computing rewards for the expected final output at intermediate denoising steps and applying gradient ascent. However, in early stages, the expected output is too blurry to provide meaningful guidance, while in later stages—when the output becomes clearer—gradient updates have minimal effect. In contrast, a one-step model generates a clear image directly from a prior sample, enabling more effective guidance. Comparative results are presented in Section 4.

**Our Technical Contribution.** The gradient ascent approach for maximizing the future reward $\hat{\mathcal{R}}$ can also be applied to the one-step model, using the following update rule on the latent $\mathbf{x}_i$ at each iteration $i$:

$$\mathbf{x}_{i+1} = \mathbf{x}_i + \eta \nabla \hat{\mathcal{R}}(\mathbf{x}_i), \tag{1}$$

where $\eta$ is a step size. However, this gradient ascent in the latent space pose several challenges: (1) the latent sample $\mathbf{x}$ may get stuck in local maxima [55, 56] before achieving the desired orientation alignment, (2) the mode-seeking nature of gradient ascent can reduce sample diversity [57], and (3) $\mathbf{x}$ may deviate from the prior latent distribution $\mathcal{N}(\mathbf{0}, \mathbf{I})$, resulting in unrealistic images [48, 58]. Although the recent reward-guided noise optimization method (ReNO [22]) employs norm-based regularization [43, 59] to enforce the latent to be close to the prior distribution, it still suffers from local optima, leading to suboptimal orientation grounding results (see comparisons in Sec. 4 and further analysis in Appendix A).

Our key idea to address this is to reformulate the problem as a *sampling problem* rather than an *optimization problem*. Specifically, we aim to sample $\mathbf{x}$ from a target distribution $q^*$ that maximizes the expected reward, while ensuring $q$ remains close to the original latent distribution:

$$q^* = \arg \max_p \mathbb{E}_{\mathbf{x} \sim p}[\hat{\mathcal{R}}(\mathbf{x})] - \alpha D_{\mathrm{KL}}(p \,\|\, q), \tag{2}$$

where $\alpha \in \mathbb{R}$ is a constant that controls the regularization strength, and $D_{\mathrm{KL}}(\cdot \,\|\, \cdot)$ is the Kullback-Leibler divergence [60]. This objective is closely related to those used in fine-tuning-based approaches for reward maximization [61, 48, 62]. However, the key difference is that, while these methods define the target distribution $q^*$ for the *output images*, we define it for the *latent samples*, setting $q$ as the prior distribution $\mathcal{N}(\mathbf{0}, \mathbf{I})$. This formulation leads to our key contribution: a novel, simple, and effective sampling method based on *Langevin dynamics*, discussed in Sec. 3.3.

Below, we first introduce our reward function designed for 3D orientation grounding (Sec. 3.2) and propose *reward-guided Langevin dynamics* to effectively sample from our target distribution $q^*$ (Sec. 3.3). Lastly, we introduce *reward-adaptive time rescaling* to speed up sampling convergence by incorporating time scheduling (Sec. 3.4).

## 3.2 Multi-Object Orientation Grounding Reward

To define our reward function $\mathcal{R}$, we first detect the described objects in the generated image using GroundingDINO [54] with the provided object phrases $\mathcal{W}$, yielding cropped object regions $\mathrm{Crop}(\mathbf{I}, w_i)$ for each phrase $w_i$. We then use OrientAnything [20] to assess how well the orientations of the detected objects align with the specified grounding conditions $\Phi$. OrientAnything represents

object orientations as categorical probability distributions over one-degree bins for each Euler angle. Let $\mathcal{D}_j(\cdot)$ denote the predicted distribution for the $j$-th angle. Following the convention from OrientAnything, we define the target distribution $\Pi(\phi_{i,j})$ as a discretized Gaussian centered at the reference angle $\phi_{i,j}$. The reward function $\mathcal{R}$ is then computed as the negative KL divergence between the predicted and target distributions, summed across all angles and all objects:

$$\mathcal{R}(\mathbf{I}, \mathcal{W}, \Phi) = -\frac{1}{N} \sum_{i=1}^{N} \sum_{j \in \mathcal{S}} D_{\mathrm{KL}}\Big(\mathcal{D}_j\big(\mathrm{Crop}(\mathbf{I}, w_i)\big) \,\Big\|\, \Pi(\phi_{i,j})\Big). \tag{3}$$

### 3.3 Reward-Guided Langevin Dynamics

We now introduce a method to effectively sample a latent $\mathbf{x}$ from our target distribution in Eq. 2. To address the limitations of vanilla gradient ascent for reward maximization (as discussed in Sec. 3.1), we propose enhancing the exploration of the sampling space of $\mathbf{x}$ by injecting *stochasticity*, which is known to help avoid local optima or saddle points [55, 56]. In particular, we show that simulating Langevin dynamics, in which the drift term is determined by our reward function (Sec. 3.2), can sample $\mathbf{x}$ that effectively aligns with the grounding orientation conditions.

**Proposition 1.** *Reward-Guided Langevin Dynamics. Let $q = \mathcal{N}(\mathbf{0}, \mathbf{I})$ denote the prior distribution, $\hat{\mathcal{R}}(\mathbf{x})$ be the pullback of a differentiable reward function, and $\mathbf{w}_t$ denote the standard Wiener process. As $t \to \infty$, the stationary distribution of the following Langevin dynamics*

$$d\mathbf{x}_t = \left(\frac{-\mathbf{x}_t + \frac{1}{\alpha}\nabla\hat{\mathcal{R}}(\mathbf{x}_t)}{2}\right) dt + d\mathbf{w}_t \tag{4}$$

*coincides with the optimal distribution of Eq. 2.*

*Proof.* See Appendix B. □

This demonstrates that simulating the Langevin stochastic differential equation (SDE) (Eq. 4) in the latent space ensures samples $\mathbf{x}$ are drawn from the target distribution $q^*$, balancing reward maximization with proximity to the prior distribution (Eq. 2). Using Euler–Maruyama discretization [63], we express its discrete-time approximation as:

$$\mathbf{x}_{i+1} \approx \mathbf{x}_i + \left(\frac{-\mathbf{x}_i + \frac{1}{\alpha}\nabla\hat{\mathcal{R}}(\mathbf{x}_i)}{2}\right) \delta t + \sqrt{\delta t}\epsilon_i, \tag{5}$$

where $\epsilon_i \sim \mathcal{N}(\mathbf{0}, \mathbf{I})$. By introducing the substitutions $\gamma \leftarrow \delta t$ and $\eta \leftarrow \frac{1}{2\alpha}$, the above expression (Eq. 5) simplifies to the following intuitive update rule:

$$\mathbf{x}_{i+1} = \sqrt{1 - \gamma}\,\mathbf{x}_i + \gamma\eta\nabla\hat{\mathcal{R}}(\mathbf{x}_i) + \sqrt{\gamma}\epsilon_i. \tag{6}$$

Notably, this final update step (Eq. 6) is *surprisingly simple*. Compared to the update step in standard gradient ascent (Eq. 1), only the stochastic noise term and scaling factor are additionally introduced, which require *just a single additional line of code*. This update rule integrates exploration through noise while explicitly ensuring proximity to a prior distribution.

### 3.4 Reward-Adaptive Time Rescaling

While our reward-guided Langevin dynamics already enables effective sampling of $\mathbf{x}$ for 3D orientation grounding, we additionally introduce *reward-adaptive time rescaling* to further accelerate convergence via time rescaling. In Leroy *et al.* [64], a time-rescaled SDE is introduced with a *monitor function* $\mathcal{G}$ that adaptively controls the step size, along with a correction drift term $\frac{1}{2}\nabla\mathcal{G}(\hat{\mathcal{R}}(\mathbf{x}))dt$ to preserve the stationary distribution of the original SDE. By defining a new time variable $\tau$ and the corresponding process $\tilde{\mathbf{x}}_\tau = \mathbf{x}_{t(\tau)}$ with correction drift term $\frac{1}{2}\nabla\mathcal{G}(\hat{\mathcal{R}}(\tilde{\mathbf{x}}_\tau))d\tau$, our time-rescaled version of reward-guided Langevin SDE in Eq. 4 is expressed as:

$$d\tilde{\mathbf{x}}_\tau = \mathcal{G}(\hat{\mathcal{R}}(\tilde{\mathbf{x}}_\tau))\left(-\frac{1}{2}\tilde{\mathbf{x}}_\tau + \eta\nabla\hat{\mathcal{R}}(\tilde{\mathbf{x}}_\tau)\right) d\tau + \frac{1}{2}\nabla\mathcal{G}(\hat{\mathcal{R}}(\tilde{\mathbf{x}}_\tau))d\tau + \sqrt{\mathcal{G}(\hat{\mathcal{R}}(\tilde{\mathbf{x}}_\tau))}d\tilde{\mathbf{w}}_\tau, \tag{7}$$

where the original time increment is rescaled according to $dt = \mathcal{G}(\hat{\mathcal{R}}(\tilde{\mathbf{x}}_\tau))d\tau$ and $d\mathbf{w}_t =$

**Algorithm 1** ORIGEN

**Require:** $c$ (Prompt), $\mathcal{W}$ (Object phrase set), $\Phi$ (Grounding orientation set), $\mathcal{F}_\theta$ (One-step T2I model), $\mathcal{R}$ (Reward), $\mathcal{G}$ (Monitor), $M$ (# Steps), $\eta$ (Scale), $\gamma$ (Step size)
1: Initialize $\mathbf{x}_0 \sim \mathcal{N}(\mathbf{0}, \mathbf{I})$, $\mathcal{R}_* = -\inf$
2: **for** $i = 0$ to $M - 1$ **do**
3:     $\hat{\mathcal{R}}_i = \mathcal{R}(\mathcal{F}_\theta(\mathbf{x}_i, c), \mathcal{W}, \Phi)$     ▷ *Reward Calc.*
4:     $\gamma_i = \mathcal{G}(\hat{\mathcal{R}}_i)\gamma$     ▷ *Timestep rescaling*
5:     $\epsilon_i \sim \mathcal{N}(\mathbf{0}, \mathbf{I})$
6:     $\mathbf{x}_{i+1} = \sqrt{1 - \gamma_i}\mathbf{x}_i + \gamma_i\eta\nabla\hat{\mathcal{R}}_i + \frac{1}{2}\gamma_i\nabla\log\mathcal{G}(\hat{\mathcal{R}}_i)$
        $+ \sqrt{\gamma_i}\epsilon_i$     ▷ *Update step*
7:     **if** $\hat{\mathcal{R}}_i > \mathcal{R}_*$ **then**
8:         $\mathcal{R}_* = \hat{\mathcal{R}}_i, \mathbf{x}_* = \mathbf{x}_i$
9:     **end if**
10: **end for**
11: **return** $\mathcal{F}_\theta(\mathbf{x}_*, c)$

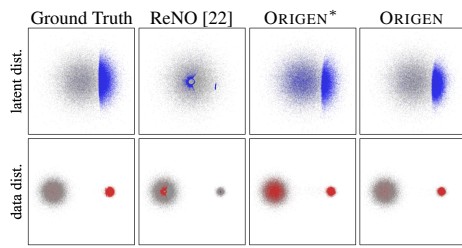

Figure 2: **Toy experiment results.** Top: latent space samples (blue); bottom: data space samples (red). Gray dots show the original distribution without reward guidance. From left to right: (1) ground truth target distribution from Eq. 2, (2) results of ReNO [22], (3) results of ours with uniform time scaling, and (4) results of ours with reward-adaptive time rescaling.

$\sqrt{\mathcal{G}(\hat{\mathcal{R}}(\tilde{\mathbf{x}}_\tau))}d\tilde{\mathbf{w}}_\tau$. A detailed derivation and convergence analysis of Eq. 7 are provided in Appendix C. Defining $\gamma(\tilde{\mathbf{x}}_\tau) = \mathcal{G}(\hat{\mathcal{R}}(\tilde{\mathbf{x}}_\tau))d\tau$, we obtain the following time-rescaled update rule via Euler-Maruyama discretization:

$$\mathbf{x}_{i+1} = \sqrt{1 - \gamma(\mathbf{x}_i)}\mathbf{x}_i + \gamma(\mathbf{x}_i)\eta\,\nabla\hat{\mathcal{R}}(\mathbf{x}_i) + \frac{1}{2}\gamma(\mathbf{x}_i)\nabla\log\mathcal{G}(\hat{\mathcal{R}}(\mathbf{x}_i)) + \sqrt{\gamma(\mathbf{x}_i)}\,\epsilon_i. \quad (8)$$

Regarding the design of the monitor function, existing work [64] suggests setting the step size inversely proportional to the squared norm of the drift coefficients in Eq. 4. However, this approach requires computing the Hessian of the reward function for the correction term, which is computationally heavy. Alternatively, we propose following simple, reward-adaptive monitor function:

$$\mathcal{G}(\hat{\mathcal{R}}(\mathbf{x})) = s_{\min} - \tanh(k\hat{\mathcal{R}}(\mathbf{x})) \cdot (s_{\max} - s_{\min}), \quad (9)$$

where the hyperparmeters $s_{\min}$, $s_{\max}$, and $k$ are set to $\frac{1}{3}$, $\frac{4}{3}$, and 0.3 in our experiments, respectively. This function adaptively scales the step size by assigning smaller steps in high-reward and larger steps in low-reward, thereby improving convergence speed and accuracy. Please refer to Fig. 5 in Appendix that provides the visualization of our monitor function $\mathcal{G}(\hat{\mathcal{R}}(\mathbf{x}))$.

In Fig. 2, we present a toy experiment demonstrating the effectiveness of our Langevin dynamics and reward-adaptive time rescaling compared to gradient ascent with regularization (ReNO [22]). See Appendix D for details on the toy experiment setup. The top row shows the ground truth target latent distribution $q^*$ (leftmost) alongside latent samples generated by different methods, and the bottom row displays the corresponding data distributions. While ReNO [22] fails to accurately capture the target distribution due to its mode-seeking behavior (2nd column, as discussed in Sec. 3.1), our method successfully aligns with it (3rd column), and time rescaling further accelerates convergence (4th column). Our sampling procedure is outlined in Alg. 1. Note that setting $\mathcal{G}(\hat{\mathcal{R}}(\mathbf{x})) = 1$ reverts the method to the uniformly time-rescaled form.

## 4 Experiments

### 4.1 Datasets

To the best of our knowledge, no existing benchmark has been proposed to evaluate 3D orientation grounding in text-to-image generation, aside from user studies conducted by Cheng *et al.* [18]. To address this, we introduce ORIBENCH based on the MS-COCO dataset [23], that consist of diverse text prompts, image, and bounding boxes.

**ORIBENCH-Single.** For a comparison with previous orientation-grounding methods [18, 17] that can condition on only a single object, we construct the ORIBENCH-Single dataset. From MS-COCO validation set [23], we filter out (1) object classes for which orientation cannot be clearly defined (e.g., objects with front-back symmetry) and (2) image captions lacking explicit object references. Since

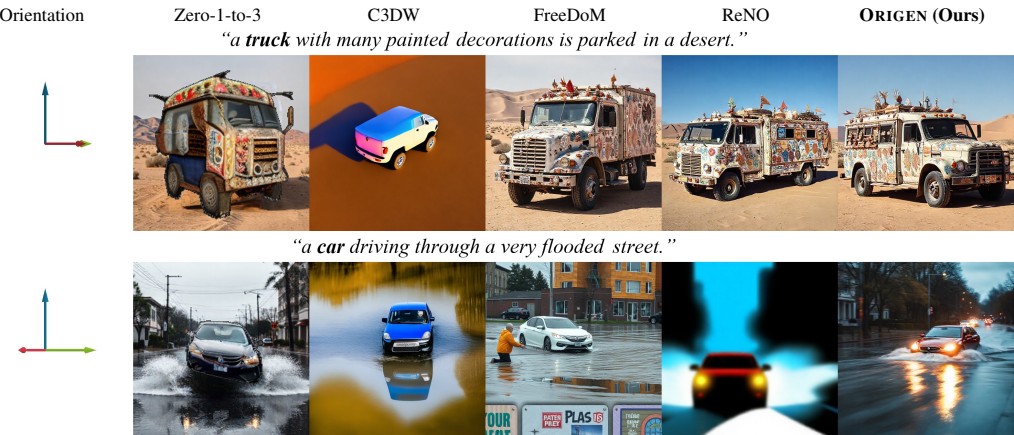

Figure 3: **Qualitative comparisons on ORIBENCH-Single benchmark (Sec. 4.5).** Compared to the existing orientation-to-image models [18, 17], ORIGEN generates the most realistic images, which also best align with the grounding conditions in the leftmost column.

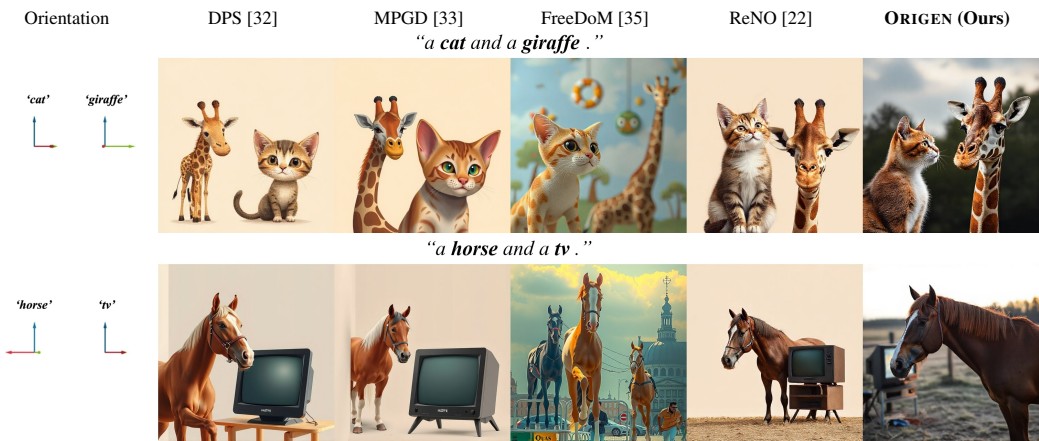

Figure 4: **Qualitative comparisons on ORIBENCH-Multi benchmark (Sec. 4.5).** Compared to the guided-generation methods [32, 33, 35, 22], ORIGEN generates the most realistic images, which also best align with the grounding conditions in the leftmost column.

the current orientation-to-image generation model [18] is only capable of controlling the front 180° range of azimuths, we further filter the samples to only include those within this range. This procedure yields 252 text-image pairs covering 25 distinct object classes. Using the provided bounding boxes, we cropped the foreground objects and fed them into OrientAnything [20], an orientation estimation model, to obtain pseudo-GT grounding orientations. Building upon this dataset, ORIBENCH-Single was constructed by mix-matching the image captions and grounding orientations, ultimately forming a dataset consisting of 25 object classes, each with 40 samples, totaling 1K samples (See Appendix E to check object classes we used).

**ORIBENCH-Multi.** For the comparison of a complex scenario with the grounding orientation of multiple objects, we construct ORIBENCH-Multi following an approach to that in ORIBENCH-Single. Since our base dataset [23] lacks samples composed solely of objects with a clear orientation, we mix-match object classes and grounding orientations from the 252 non-overlapping text-image pairs in ORIBENCH-Single, forming a dataset consisting of 371 samples, each containing a varying number of objects. We annotated prompts by concatenating individual object phrases (e.g., "a cat, and a dog.").

## 4.2 Evaluation Metrics

We compare ORIGEN's performance from two perspectives: (1) Orientation Alignment and (2) Text-to-Image Alignment. We measure orientation alignment using two metrics: 1) Acc.@22.5°,

Table 1: **Quantitative comparisons on 3D orientation grounded image generation.** Best and second-best results are highlighted in **bold** and underlined, respectively. ORIGEN* denotes ours without reward-adaptive time rescaling.

| Id | Model | ORIBENCH-Single | | | | | ORIBENCH-Multi | | | | |
|---|---|---|---|---|---|---|---|---|---|---|---|
| | | Orientation Alignment | | Text-to-Image Alignment | | | Orientation Alignment | | Text-to-Image Alignment | | |
| | | Acc.@22.5°↑ | Abs. Err.↓ | CLIP↑ | VQA↑ | PickScore↑ | Acc.@22.5°↑ | Abs. Err.↓ | CLIP↑ | VQA↑ | PickScore↑ |
| | | (1) Fine-tuned Orientation-to-Image Model | | | | | | | | | |
| 1 | Zero-1-to-3 [17] | 0.499 | 59.03 | **0.272** | 0.663 | 0.213 | – | – | – | – | – |
| 2 | C3DW [18] | 0.426 | 64.77 | 0.220 | 0.439 | 0.197 | – | – | – | – | – |
| | | (2) Guided-Generation Methods with Multi-Step Model | | | | | | | | | |
| 3 | DPS [32] | 0.664 | 28.16 | 0.246 | 0.662 | 0.221 | 0.487 | 38.28 | 0.285 | 0.742 | 0.231 |
| 4 | MPGD [33] | 0.689 | 27.62 | 0.246 | 0.661 | 0.222 | 0.532 | 35.03 | 0.283 | 0.739 | **0.232** |
| 5 | FreeDoM [35] | 0.741 | 20.90 | 0.259 | 0.728 | **0.225** | 0.591 | 31.77 | 0.279 | 0.783 | 0.227 |
| | | (3) Guided-Generation Methods with One-Step Model | | | | | | | | | |
| 6 | ReNO [22] | 0.796 | 20.56 | 0.247 | 0.663 | 0.212 | 0.478 | 45.95 | 0.277 | 0.756 | 0.223 |
| 7 | ORIGEN* (Ours) | 0.854 | 18.28 | 0.265 | 0.732 | 0.224 | 0.679 | 29.7 | 0.287 | 0.804 | 0.225 |
| 8 | ORIGEN (Ours) | **0.871** | **17.41** | 0.265 | **0.735** | 0.224 | **0.692** | **28.4** | 0.287 | **0.807** | 0.225 |

the angular accuracy within a tolerance of $\pm 22.5°$ and 2) the absolute error on azimuth angles[1] between the predicted and grounding object orientations, following Wang *et al.* [20]. For evaluation, we use OrientAnything [20] to predict the 3D orientation from the generated images. Since its predictions may not be perfect, we also conduct a user study in Sec. 4.6 to validate the results. Along with grounding accuracy, we also evaluate text-to-image alignment using CLIP Score [65], VQA-Score [66], and PickScore [67].

### 4.3  Baselines

We compare ORIGEN with three types of baselines: (1) training-based orientation-to-image generation methods [18, 17] (2) training-free guided generation methods [35, 22] for the multi-step generation, and (3) training-free guided generation methods for the one-step generation. For (1), we include Continuous 3D Words (C3DW) [18] and Zero-1-to-3 [17], following the baselines used in the most recent work in this field [18]. For (2), we consider DPS [32], MPGD [33], and FreeDoM [35], which update the intermediate samples at each step of the multi-step sampling process based on the expected reward computed on the estimated clean image. Finally, for (3), we compare ORIGEN with ReNO [22], which optimizes the initial latent through vanilla gradient ascent using one-step models. Furthermore, to evaluate the effectiveness of our reward adaptive time rescaling, we perform additional comparison with our method variant without this component, denoted as ORIGEN*.

### 4.4  Implementation Details

We use FLUX-Schnell [21] as the one-step generative model for both ORIGEN and ReNO [22], while all multi-step guided generation baselines (DPS [32], MPGD [33], and FreeDoM [35]) are implemented using FLUX-Dev [21] as the multi-step generative model. Except for the training-based methods (C3DW [18] and Zero-1-to-3 [17]), we match the number of function evaluations (NFEs—defined as the number of iterations in our method and the denoising steps in multi-step generative models) to 50 for a fair comparison across all methods. All experiments were conducted on a single NVIDIA 48GB VRAM A6000 GPU. Further implementation details are provided in Appendix F.

### 4.5  Results

**ORIBENCH-Single.**   In the left part of Tab. 1, we show our quantitative comparison results on the ORIBENCH-Single benchmark. ORIGEN *significantly* outperforms all the baselines in orientation alignment, showing comparable performance in text-to-image alignment. Our qualitative comparisons are also shown in Fig. 3, demonstrating that ORIGEN generates high-quality images that align with the orientation conditions and input text prompts. Note that C3DW [18] is trained on synthetic data (i.e., multi-view renderings of a 3D object) to learn orientation-to-image generation. Thus, it has limited generalizability to real-world images and the output images lack realism. Zero-1-to-3 [17] is also trained on single-object images but *without backgrounds*, requiring additional

---

[1]We perform comparisons only on azimuth angle, as existing methods [18, 17] do not support the control over polar and rotation angles. Note that our results for all azimuth, polar, and rotation angles are provided in Appendix G.1.

Table 2: **User Study Results.** 3D orientation-grounded text-to-image generation results of ORIGEN was preferred by 58.18% of the participants on Amazon Mechanical Turk [69], significantly outperforming the baselines [18, 17].

| Method | Zero-1-to-3 [17] | C3DW [18] | ORIGEN (Ours) |
|---|---|---|---|
| User Preferences ↑ (%) | 20.58 | 21.24 | **58.18** |

background image composition (also used in the evaluation of C3DW [18]) that may introduce unnatrual artifacts. The existing methods on guided generation methods also achieve suboptimal results compared to ORIGEN. In particular, all training-free multi-step guidance methods, including DPS [32], MPGD [33], and FreeDoM [35], perform poorly in orientation grounding. This is because object orientation control relies on modifying low-frequency image structures, which are primarily formed during early sampling stages, where multi-step diffusion models produce noisy and blurry outputs [68] (as discussed in Sec. 3.1). ReNO [22] also achieves suboptimal results compared to ours, as it performs latent optimization based on vanilla gradient ascent which is prone to local optima (as discussed in Sec. 3.1). Overall, our method achieves the best results both with and without time rescaling, demonstrating its ability to effectively maximize rewards while avoiding over-optimization.

**ORIBENCH-Multi.** We additionally show the quantitative comparison results on the ORIBENCH-Multi benchmark. Since no fine-tuned methods ((1) in Tab. 1) are capable of multi-object orientation grounding, we assess our method only with guided generation methods ((2), (3) in Tab. 1). Our quantitative results are presented in the right of Tab. 1, and qualitative examples are provided in Fig. 4. As shown, ORIGEN consistently outperforms all guided generation baselines under the same reward function. These results demonstrate that our object-agnostic reward design, combined with the proposed method, enables robust and generalizable orientation grounding even in complex multi-object scenarios.

**Additional Results.** We provide additional results for orientation grounding task (1) under more diverse orientations, and (2) for four primitive views (front, left, right, back) to enable comparisons with text-to-image models, where the orientation condition is included as an input text prompt. Please refer to Appendix G.

### 4.6 User Study

Our previous quantitative evaluations were performed by comparing the grounding orientations and orientations *estimated* from the generated images using OrientAnything [20]. While its orientation estimation performance is highly robust (as seen in all of our qualitative results), we additionally conduct a user study to further validate the effectiveness of our method based on human evaluation performed by 100 participants on Amazon Mechanical Turk [69]. Each participant was presented with the grounding orientation, the input prompt, and the images generated by (1) Zero-1-to-3 [17], (2) C3DW [18], and (3) ORIGEN. Since ORIGEN outperformed all guided-generation methods with same reward function design in quantitative comparison, they were excluded in the user study. Then, participants were asked to select the image that best matches both the grounding orientations and the input text prompt – directly following the user study settings in C3DW [18]. In Tab. 2, ORIGEN was preferred by 58.18% of the participants, clearly outperforming the baselines. For more details of this user study, refer to Appendix H.

### 4.7 Computation Time

In our experiments, we set the number of iterations in our method to match the denoising steps in multi-step generative models (e.g., 50), resulting in comparable computation time—approximately 52.7 seconds per image. A detailed comparison is provided in Appendix I. Notably, for cases achieving angular accuracy within a tolerance of $\pm 22.5°$ (Tab. 3), the average number of iterations to reach this threshold was 12.8, corresponding to an average of 13.5 seconds. Our approach can also serve as a post-refinement method for existing models, potentially further reducing computation time.

## 5 Conclusion

We presented ORIGEN, the first 3D orientation grounding method for text-to-image generation across multiple objects and categories. To enable test-time guidance with a pretrained discriminator, we introduced a Langevin-based sampling approach with reward-adaptive time rescaling for faster convergence. Experiments showed that our method outperforms fine-tuning-based baselines and uniquely supports conditioning on multiple open-vocabulary objects.

**Limitations and Societal Impact.** Since our method relies on a pretrained discriminator, its performance is inherently bounded by the quality of that discriminator—though in our case, it still achieved state-of-the-art results by a significant margin. As a generative AI technique, our method may be misused to produce inappropriate or harmful content, underscoring the importance of responsible development and deployment.

## Acknowledgement

This work was supported by the NRF of Korea (RS-2023-00209723); IITP grants (RS-2022-II220594, RS-2023-00227592, RS-2024-00399817, RS-2025-25441313, RS-2025-25443318, RS-2025-02653113); the National Program for Excellence in SW, supervised by the IITP; and the Technology Innovation Program (RS-2025-02317326), all funded by the Korean government (MSIT and MOTIE), as well as by the DRB-KAIST SketchTheFuture Research Center.

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

# Appendix

## A    Analysis on Norm-Based Regularization

In this section, we analyze the limitations of norm-based regularization in reward-guided noise optimization by highlighting its mode-seeking property, which can lead to overoptimization and convergence to local maxima. Consider the following reward maximization process with norm-based regularization [22, 46]:

$$\mathbf{x}_{t+1} = \mathbf{x}_t + \eta_1 \nabla \hat{\mathcal{R}}(\mathbf{x}_t) + \eta_2 \nabla \log y(\|\mathbf{x}_t\|), \tag{10}$$

where $\hat{\mathcal{R}} = \mathcal{R} \circ \mathcal{F}_\theta$ is the pullback of the reward function $\mathcal{R}$ and $y(\| \cdot \|)$ represents the probability density of a $\chi^d$ distribution.

We can view both $\hat{\mathcal{R}}(\cdot)$ and $y(\| \cdot \|)$ as components of a single reward, allowing us to rewrite the update as

$$
\begin{aligned}
\mathbf{x}_{t+1} &= \mathbf{x}_t + \eta_1 \nabla \hat{\mathcal{R}}(\mathbf{x}_t) + \eta_2 \nabla \log y(\|\mathbf{x}_t\|) \\
&= \mathbf{x}_t + \eta_1 \nabla \hat{\mathcal{R}}(\mathbf{x}_t) + \eta_2 \nabla \left[ (d-1) \log \|\mathbf{x}_t\|^2 - \frac{1}{2}\|\mathbf{x}_t\|^2 \right] \\
&= \mathbf{x}_t + \eta_1 \nabla \hat{\mathcal{R}}(\mathbf{x}_t) + \eta_2 \nabla K(\mathbf{x}_t) \\
&= \mathbf{x}_t + \nabla \Phi(\mathbf{x}_t)
\end{aligned}
\tag{11}
$$

where $\Phi(\mathbf{x}) = \eta_1 \hat{\mathcal{R}}(\mathbf{x}) + \eta_2 K(\mathbf{x})$. This optimization process is an Euler discretization (with $\delta t = 1$) of the ODE

$$\frac{d\mathbf{x}}{dt} = \nabla \Phi(\mathbf{x}), \tag{12}$$

which represents a deterministic gradient flow, whose stationary measure is a weighted sum of Dirac deltas located at the maximizers of $\Phi(\mathbf{x}) = \eta_1 \hat{\mathcal{R}}(\mathbf{x}) + \eta_2 K(\mathbf{x}) = 0$.

However, as illustrated in Fig. 2 in the main paper, this deterministic gradient ascent does not directly prevent the iterates from deviating substantially from the original latent distribution. Also, the process may collapse to local maxima–even ones where the reward is not sufficiently high [70]. Our empricial observations indicate that this approach is ineffective for our application, presumably because the reward function defined over the latent exhibits many local maxima, causing the deterministic ascent to converge to suboptimal solutions.

## B    Proofs

***Proof of Proposition 1***. Recall the standard result from overdamped Langevin dynamics: if $\mathbf{x}_t$ evolves according to the SDE:

$$d\mathbf{x}_t = \frac{1}{2}\nabla \log q^*(\mathbf{x}_t)dt + d\mathbf{w}_t, \tag{13}$$

then $q^*(\mathbf{x})$ is the unique stationary distribution. Using Eq. 2 in the main paper, we obtain

$$\frac{1}{2}\nabla \log q^*(\mathbf{x}) = -\frac{1}{2}\mathbf{x} + \frac{1}{2\alpha}\nabla \hat{\mathcal{R}}(\mathbf{x}_t). \tag{14}$$

Integrating this with respect to $\mathbf{x}$ gives

$$q^*(\mathbf{x}) \propto q(\mathbf{x}) \exp\left( \frac{\hat{\mathcal{R}}(\mathbf{x})}{\alpha} \right), \tag{15}$$

where $q(\mathbf{x})$ is a standard Gaussian distribution.

Finally, following existing approach [61], we easily arrive at

$$q^* = \arg\max_p \mathbb{E}_{\mathbf{x} \sim p}[\hat{\mathcal{R}}(\mathbf{x})] - \alpha D_{\mathrm{KL}}(p\|q), \tag{16}$$

which matches the expression presented in Eq. 2 in the main paper. $\qquad\square$

## C   Analysis on Reward-Adaptive Time-Rescaled SDE

In this section, we analyze the effect of a position-dependent step size on reward-guided Langevin dynamics. We first illustrate why a naive time-rescaling approach fails to preserve the desired stationary distribution, and how to fix it with an additional correction term, eventually leading to Eq. 8 in the main paper. Our derivation follows the approach of Leroy *et al.* [64].

### C.1   Non-Convergence of Direct Time-Rescaled SDE

Consider the following SDE:

$$d\mathbf{x}_t = b(\mathbf{x}_t)\,dt + \sigma(\mathbf{x}_t)\,d\mathbf{w}_t, \tag{17}$$

with drift $b(\mathbf{x}_t)$ and diffusion coefficient $\sigma(\mathbf{x}_t)$. Its probability density $\rho(\mathbf{x}, t)$ evolves according to the Fokker-Planck equation [71]:

$$\frac{\partial}{\partial t}\rho(\mathbf{x}, t) = -\nabla \cdot [b(\mathbf{x})\rho(\mathbf{x}, t)] + \frac{1}{2}\nabla^2\big[\sigma^2(\mathbf{x})\rho(\mathbf{x}, t)\big]. \tag{18}$$

Thus, the stationary distribution must lie in the *kernel* of the corresponding Fokker–Planck operator $\mathcal{L}^*$ [72]:

$$\mathcal{L}^*\rho(\mathbf{x}) = -\nabla \cdot \Big(b(\mathbf{x})\,\rho(\mathbf{x})\Big) + \frac{1}{2}\nabla^2\Big[\sigma^2(\mathbf{x})\,\rho(\mathbf{x})\Big]. \tag{19}$$

Now, consider the reward-guided Langevin SDE in Eq. 4 in the main paper:

$$d\mathbf{x}_t = \left(-\frac{1}{2}\mathbf{x}_t + \eta\,\nabla\hat{\mathcal{R}}(\mathbf{x}_t)\right)dt + d\mathbf{w}_t, \tag{20}$$

which has the stationary distribution $q^*(\mathbf{x})$ given by Eq. 2 in the main paper. By comparing with the general form, we identify:

$$b(\mathbf{x}) = -\frac{1}{2}\mathbf{x} + \eta\,\nabla\hat{\mathcal{R}}(\mathbf{x}), \quad \sigma(\mathbf{x}) = 1. \tag{21}$$

Hence,

$$\mathcal{L}^*q^*(\mathbf{x}) = -\nabla \cdot \left[\left(-\frac{1}{2}\mathbf{x} + \eta\,\nabla\hat{\mathcal{R}}(\mathbf{x})\right)q^*(\mathbf{x})\right] + \frac{1}{2}\nabla^2\Big[q^*(\mathbf{x})\Big]$$
$$= 0. \tag{22}$$

Next, we introduce a monitor function $\mathcal{G}(\mathbf{x}) > 0$ and define a new time variable $\tau$ by

$$dt = \mathcal{G}\big(\mathbf{x}_{t(\tau)}\big)\,d\tau. \tag{23}$$

Defining the time-rescaled process $\tilde{\mathbf{x}}_\tau = \mathbf{x}_{t(\tau)}$, we rewrite the SDE as

$$d\tilde{\mathbf{x}}_\tau = \mathcal{G}(\tilde{\mathbf{x}}_\tau)\left(-\frac{1}{2}\tilde{\mathbf{x}}_\tau + \eta\,\nabla\hat{\mathcal{R}}(\tilde{\mathbf{x}}_\tau)\right)d\tau + \sqrt{\mathcal{G}(\tilde{\mathbf{x}}_\tau)}\,d\tilde{W}_\tau. \tag{24}$$

In this rescaled SDE, the new drift and diffusion coefficients are

$$a(\tilde{\mathbf{x}}) = \mathcal{G}(\tilde{\mathbf{x}})\left(-\frac{1}{2}\tilde{\mathbf{x}} + \eta\,\nabla\hat{\mathcal{R}}(\tilde{\mathbf{x}})\right), \quad \sigma(\tilde{\mathbf{x}}) = \sqrt{\mathcal{G}(\tilde{\mathbf{x}})}. \tag{25}$$

Table 3: **Convergence speed to the desired success criterion.** ORIGEN$^*$ denotes ours without reward-adaptive time rescaling.

| Method | ORIGEN$^*$ | ORIGEN |
|---|---|---|
| Mean NFE $\downarrow$ | 14.2 | **12.8** |
| Inference time (s) $\downarrow$ | 14.9 | **13.5** |

Applying the corresponding Fokker–Planck operator $\tilde{\mathcal{L}}^*$ to $q^*(\mathbf{x})$, we obtain

$$\tilde{\mathcal{L}}^* q^*(\tilde{\mathbf{x}}) = -\nabla \cdot \left[ \mathcal{G}(\tilde{\mathbf{x}}) \left( -\frac{1}{2}\tilde{\mathbf{x}} + \eta \, \nabla \hat{\mathcal{R}}(\tilde{\mathbf{x}}) \right) q^*(\tilde{\mathbf{x}}) \right] + \frac{1}{2}\nabla^2 \left[ \mathcal{G}(\tilde{\mathbf{x}}) q^*(\tilde{\mathbf{x}}) \right]$$

$$= -\nabla \cdot \left[ a(\tilde{\mathbf{x}}) \, q^*(\tilde{\mathbf{x}}) \right] + \frac{1}{2}\nabla \cdot \left[ \nabla \mathcal{G}(\tilde{\mathbf{x}}) \, q^*(\tilde{\mathbf{x}}) + \mathcal{G}(\tilde{\mathbf{x}}) \, \nabla q^*(\tilde{\mathbf{x}}) \right]$$

$$= -\nabla \cdot \left[ a(\tilde{\mathbf{x}}) \, q^*(\tilde{\mathbf{x}}) \right] + \frac{1}{2}\nabla \cdot \left[ \nabla \mathcal{G}(\tilde{\mathbf{x}}) \, q^*(\tilde{\mathbf{x}}) + 2a(\tilde{\mathbf{x}}) \, q^*(\tilde{\mathbf{x}}) \right]$$

$$= \frac{1}{2}\nabla \cdot \left[ \nabla \mathcal{G}(\tilde{\mathbf{x}}) \, q^*(\tilde{\mathbf{x}}) \right]$$

$$\neq 0, \tag{26}$$

where we used the fact that $\mathcal{G}(\mathbf{x}) \, \nabla q^*(\mathbf{x}) = 2a(\mathbf{x}) \, q^*(\mathbf{x})$, which follows from the identity $\nabla \log q^*(\mathbf{x}) = -\mathbf{x} + 2\eta \, \nabla \hat{\mathcal{R}}(\mathbf{x})$. Therefore, $q^*(\mathbf{x})$ is not annihilated by $\tilde{\mathcal{L}}^*$, implying that the time-rescaled SDE does not converge to the desired target distribution in Eq. 2 in the main paper.

## C.2 Time-Rescaled SDE with Correction Drift Term

**Convergence guarantee to original stationary distribution.** Following previous work [64], we can add a correction drift term $\frac{1}{2}\nabla \mathcal{G}(\tilde{\mathbf{x}}_\tau)$ to Eq. 24 to ensure that the rescaled process converges to the same stationary distribution as the original SDE. The modified SDE becomes

$$d\tilde{\mathbf{x}}_\tau = \mathcal{G}(\tilde{\mathbf{x}}_\tau) \left( -\frac{1}{2}\tilde{\mathbf{x}}_\tau + \eta \, \nabla \hat{\mathcal{R}}(\tilde{\mathbf{x}}_\tau) \right) d\tau + \frac{1}{2}\nabla \mathcal{G}(\tilde{\mathbf{x}}_\tau) d\tau + \sqrt{\mathcal{G}(\tilde{\mathbf{x}}_\tau)} \, d\tilde{W}_\tau. \tag{27}$$

The corresponding Fokker-Planck operator $\tilde{\mathcal{L}}^*_{\text{corr}}$ for this corrected time-rescaled SDE now annihilates the original stationary distribution $q^*(\mathbf{x})$:

$$\tilde{\mathcal{L}}^*_{\text{corr}} q^*(\tilde{\mathbf{x}}) = -\nabla \cdot \left[ \mathcal{G}(\tilde{\mathbf{x}}) \left( -\frac{1}{2}\tilde{\mathbf{x}} + \eta \, \nabla \hat{\mathcal{R}}(\tilde{\mathbf{x}}) \right) q^*(\tilde{\mathbf{x}}) + \frac{1}{2}\nabla \mathcal{G}(\tilde{\mathbf{x}}) \, q^*(\tilde{\mathbf{x}}) \right] + \frac{1}{2}\nabla^2 \left[ \mathcal{G}(\tilde{\mathbf{x}}) q^*(\tilde{\mathbf{x}}) \right]$$

$$= \frac{1}{2}\nabla \cdot \left[ \nabla \mathcal{G}(\tilde{\mathbf{x}}) \, q^*(\tilde{\mathbf{x}}) \right] - \frac{1}{2}\nabla \cdot \left[ \nabla \mathcal{G}(\tilde{\mathbf{x}}) \, q^*(\tilde{\mathbf{x}}) \right]$$

$$= 0. \tag{28}$$

Consequently $q^*(\tilde{\mathbf{x}})$ remains in the kernel of the $\tilde{\mathcal{L}}^*_{\text{corr}}$, showing that this corrected time-rescaled SDE preserves the original invariant distribution.

**Convergence speed of time rescaling approach.** To verify the effectiveness of our time-rescaling approach, we analyzed the average number of iterations required to reach the desired success criterion during reward-guided Langevin dynamics on ORIBENCH-Single dataset. As shown in Tab. 3, ORIGEN$^*$ requires an average of *14.2* iterations to reach the desired reward level, whereas ORIGEN reduced this to *12.8* iterations, demonstrating improved convergence speed without image quality degradation. Notably, the computational overhead introduced by adaptive rescaling is negligible, as the term $\nabla \log \mathcal{G}(\hat{\mathcal{R}}_i)$ in Alg. 1 in the main paper can be efficiently computed via the chain rule, using the precomputed reward gradient $\nabla \hat{\mathcal{R}}_i$.

# D   Setup for the Toy Experiment

We train a *rectified flow model* [41] on a 2D domain, where the source distribution is $\mathcal{N}(0, \mathbf{I})$ and the target distribution is a mixture of two Gaussians, $\mathcal{N}(\mu_1, \sigma_1 \mathbf{I})$ and $\mathcal{N}(\mu_2, \sigma_2 \mathbf{I})$, with $\mu_1 =$

| Class Number | Class Name |
|:---:|:---|
| 1 | Airplane |
| 2 | Bear |
| 3 | Bench |
| 4 | Bicycle |
| 5 | Bird |
| 6 | Boat |
| 7 | Bus |
| 8 | Car |
| 9 | Cat |
| 10 | Chair |
| 11 | Cow |
| 12 | Dog |
| 13 | Elephant |
| 14 | Giraffe |
| 15 | Horse |
| 16 | Laptop |
| 17 | Motorcycle |
| 18 | Person |
| 19 | Sheep |
| 20 | Teddy bear |
| 21 | Toilet |
| 22 | Train |
| 23 | Truck |
| 24 | Tv |
| 25 | Zebra |

Table 4: **Selected object classes in our dataset.**

$(4, 0)^T, \mu_2 = (-4, 0)^T, \sigma_1 = 0.3$, and $\sigma_2 = 0.9$. The velocity prediction network consists of four hidden MLP layers of width $128$. We first train an initial model and distill it twice, resulting in a 3-rectified flow model. The reward function is defined as

$$\mathcal{R}(x, y) = \exp\left(-\frac{(x-4)^2 + y^2}{2}\right) + 0.1 \cdot \exp\left(-\frac{(x+4)^2 + y^2}{2}\right) - 1. \tag{29}$$

In Fig. 2 in the main paper, all methods use the same total number of sampling steps.

## E   Selected Classes

When constructing the ORIBENCH benchmark, we selected a total of 25 classes from the 80 object classes in MS-COCO validation set [23]. We excluded classes for which distinguishing the front and back is difficult or where defining a canonical orientation is ambiguous. The remaining 25 classes were chosen based on their clear orientation cues. The detailed list of selected classes is provided in Tab. 4.

## F   Implementation Details

Following the convention of OrientAnything [20], we set the standard deviation hyperparameters for the azimuth, polar, and rotation distributions to 20, 2, and 1, respectively, to transform discrete angles into the orientation probability distribution $\Pi$. For image quality evaluation, we utilized the official implementation of VQA-Score [66], and assessed CLIP Score [65], VQA-Score [66], and Pickscore [67] using OpenAI's CLIP-ViT-L-14-336 [73], LLaVA-v1.5-13b [74], and Pickscore-v1 [67] models, respectively.

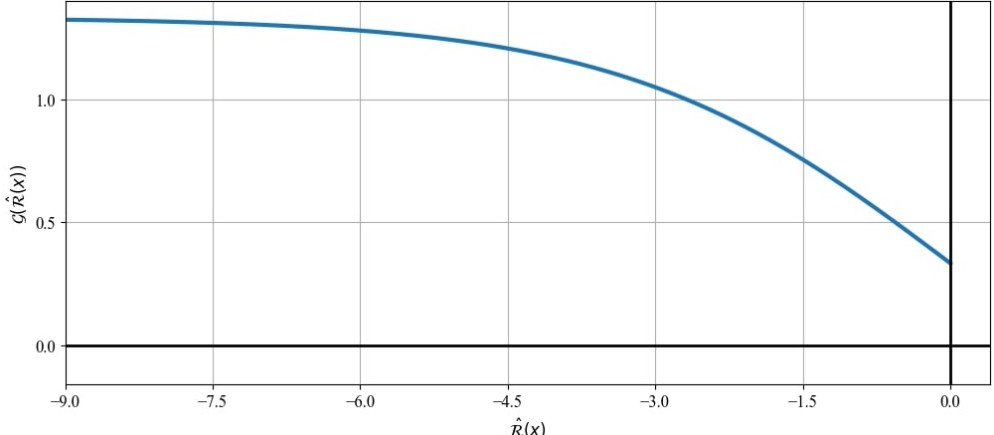

Figure 5: **Plot of our monitor function** $\mathcal{G}(\hat{\mathcal{R}}(\mathbf{x}))$.

**About ORIGEN.** We employed FLUX-Schnell [21] as our *one-step* T2I generative model and OrientAnything (ViT-L) [20] to measure the Orientation Grounding Reward, as detailed in Sec. 3.2. For all experiments, we set $\eta = 0.8$ in Alg. 1, and used $\gamma = 0.3$ for ORIBENCH-Single and $\gamma = 0.2$ for ORIBENCH-Multi, as this configuration provides a favorable balance between image quality and computational cost when evaluating with 50 NFEs. As described in Sec. 3.4, we additionally visualize our monitor function $\mathcal{G}(\hat{\mathcal{R}}(\mathbf{x}))$ from Eq. 9 in Fig. 5, which demonstrates how the function adaptively scales the step size–assigning smaller steps in high-reward regions and larger steps in low-reward regions–thereby improving convergence speed.

**About Guided Generation Methods.** For ReNO [22], we employed FLUX-Schnell [21] as the base one-step text-to-image generative model and OrientAnything (ViT-L) [20] as in ours. For multi-step guidance methods, we used FLUX-Dev [21] as multi-step text-to-image generative model. We set the gradient weight hyperparameter to 0.3 for ORIBENCH-Single and 0.2 for ORIBENCH-Multi for all methods. In the case of ReNO [22], we use norm-based regularization weight of 0.01. To ensure a fair comparison, we matched the number of function evaluations (NFEs) for all training-free one-step, multi-step guidance methods to 50.

**About Zero-1-to-3 [17].** For the Zero-1-to-3 [17] baseline, we generated the base image using FLUX-Schnell (512×512) with 4 steps and encouraged the model to generate front-facing objects by adding the "facing front" prompt. The model was then conditioned on the target azimuth while keeping the polar angle fixed at 0° to generate novel foreground views, which were later composited with the background. The foreground was segmented using SAM [75], and missing background pixels were inpainted using LaMa [76] before composition.

**About C3DW [18].** For C3DW [18], we utilized the orientation & illumination model checkpoint provided in the official implementation. This model is only capable of controlling azimuth angles for half-front views and was trained with scalar orientation values ranging from 0.0 to 0.5, where 0.0 corresponds to 90°, and 0.5 corresponds to -90°, with intermediate orientations obtained through linear interpolation. Using this mapping, we converted ground truth (GT) orientations into the corresponding input values for the model.

**Handling Back-Facing Generation.** We empirically observed that our base model struggled to generate back-facing images when the prompt did not explicitly contain view information. We hypothesized that this issue arises because high-reward samples lie within an extremely sparse region of the conditional probability space, making them inherently challenging to sample, even when employing our proposed approach. To address this issue, we explicitly added the phrase "facing back" in facing back cases (i.e., where $90 < \phi_i^{az} < 270$) and optimized the noise accordingly. With this simple adjustment, the model effectively generated both facing front and back images while maintaining high-quality outputs.

Table 5: **Quantitative comparisons on General Orientation Controllability.** ORIGEN maintains high accuracy even when evaluated on a more general set of samples. Best and second-best results are highlighted in **bold** and underlined, respectively.

| Id | Model | Orientation Alignment | | | | | | Text-to-Image Alignment | | |
|---|---|---|---|---|---|---|---|---|---|---|
| | | Azi, Acc.
@22.5° ↑ | Azi, Abs.
Err. ↓ | Pol, Acc.
@5° ↑ | Pol, Abs.
Err. ↓ | Rot, Acc.
@5° ↑ | Rot, Abs.
Err. ↓ | CLIP ↑ | VQA ↑ | Pick
Score ↑ |
| 1 | DPS [32] | 0.432 | 43.89 | 0.511 | 13.07 | 0.969 | 1.66 | 0.259 | 0.704 | **0.234** |
| 2 | MPGD [33] | 0.398 | 48.09 | 0.485 | 12.25 | 0.967 | 1.79 | 0.258 | 0.707 | **0.234** |
| 3 | FreeDoM [35] | 0.470 | 38.09 | 0.554 | 12.71 | **0.971** | **1.42** | 0.261 | 0.709 | 0.229 |
| 4 | ReNO [22] | 0.586 | 41.41 | 0.502 | 14.37 | 0.958 | 2.69 | 0.253 | 0.676 | 0.216 |
| 5 | **ORIGEN (Ours)** | **0.777** | **24.96** | **0.575** | **12.46** | 0.969 | 1.52 | **0.263** | **0.710** | 0.219 |

Table 6: **Quantitative comparisons on 3D orientation grounded image generation for four primitive views.** Best and second-best results are highlighted in **bold** and underlined, respectively.

| Id | Model | 3-View (front, left, right) | | | | | 4-View (front, left, right, back) | | | | |
|---|---|---|---|---|---|---|---|---|---|---|---|
| | | Orientation Alignment | | Text-to-Image Alignment | | | Orientation Alignment | | Text-to-Image Alignment | | |
| | | Acc.@22.5° ↑ | Abs. Err. ↓ | CLIP ↑ | VQA ↑ | PickScore ↑ | Acc.@22.5° ↑ | Abs. Err. ↓ | CLIP ↑ | VQA ↑ | PickScore ↑ |
| | | (1) One-step Text-to-Image Model | | | | | | | | | |
| 1 | SD-Turbo [24] | 0.257 | 75.09 | 0.261 | 0.721 | 0.223 | 0.244 | 78.47 | 0.262 | 0.717 | 0.223 |
| 2 | SDXL-Turbo [25] | 0.189 | 78.44 | 0.265 | 0.722 | 0.227 | 0.196 | 81.88 | 0.262 | 0.717 | 0.227 |
| 3 | FLUX-Schnell [21] | 0.312 | 75.04 | **0.268** | **0.739** | **0.230** | 0.424 | 60.26 | **0.267** | **0.739** | **0.229** |
| | | (2) Fine-tuned Orientation-to-Image Model | | | | | | | | | |
| 4 | Zero-1-to-3 [17] | 0.366 | 59.03 | 0.266 | 0.646 | 0.210 | 0.321 | 75.10 | 0.264 | 0.642 | 0.209 |
| 5 | C3DW [18] | 0.504 | 64.77 | 0.187 | 0.334 | 0.188 | – | – | – | – | – |
| | | (3) Guided-Generation Methods with One-Step Model | | | | | | | | | |
| 6 | **ORIGEN (Ours)** | **0.824** | **20.99** | 0.262 | 0.721 | 0.220 | **0.866** | **17.45** | 0.262 | 0.720 | 0.220 |

# G   Additional Results on the Extended ORIBENCH-Single Benchmark

We further evaluate ORIGEN under two additional scenarios that better reflect real-world use cases, by extending the ORIBENCH-Single Benchmark. The first experiment investigates the question: (1) *"Can* ORIGEN *handle more general and complex orientations beyond simple angles?"*. The second experiment addresses: (2) *"Can orientation grounding be achieved simply by prompting a text-to-image generative model?"*. For the first scenario, we evaluate ORIGEN and other guided generation methods on an extended benchmark covering all three orientation components—azimuth, polar, and rotation without filtering the front range of azimuths. For the second scenario, we assess the capability of text-to-image models to perform orientation grounding for four primitive directions (front, left, right, back), where the orientation condition is provided via the input text prompt.

## G.1   Orientation Grounding in more general and complex scenario.

We provide our extensive evaluation on general curated ORIBENCH-Single dataset. As discussed in Sec. 4.1, we filter out non-clear object classes and image captions, but we do not apply filtering on the front range and do not fix the polar and rotation angles. Upon this, we mix-match object classes and grounding orientations, forming a dataset consisting of 25 object classes, each with 40 samples, totaling 1K samples. We evaluated on same metrics as in Sec. 4.2 in the main paper, including polar and rotation accuracy, within a tolerance of ±5.0° as well as their absolute errors, following Wang *et al.* [20]. As shown in Tab. 5, ORIGEN generalizes well on diverse orientation conditions, outperforming all other training-free guidance methods. We also present qualitative comparisons for other training-free guidance methods in Fig. 6 . ORIGEN shows more consistent and accurate orientation alignment compared to other training-free approaches. This demonstrates the robustness of our approach, maintaining high orientation alignment performance even when evaluated on a more general and complex set of samples.

## G.2   Text-to-Image Models.

As baselines, we consider several one-step T2I generative models: SD-Turbo [24], SDXL-Turbo [25], and FLUX-Schnell [21]. In particular, we appended a phrase that specifies the object orientation

| Orientation | DPS [32] | MPGD [33] | FreeDoM [35] | ReNO [22] | ORIGEN (Ours) |
|---|---|---|---|---|---|

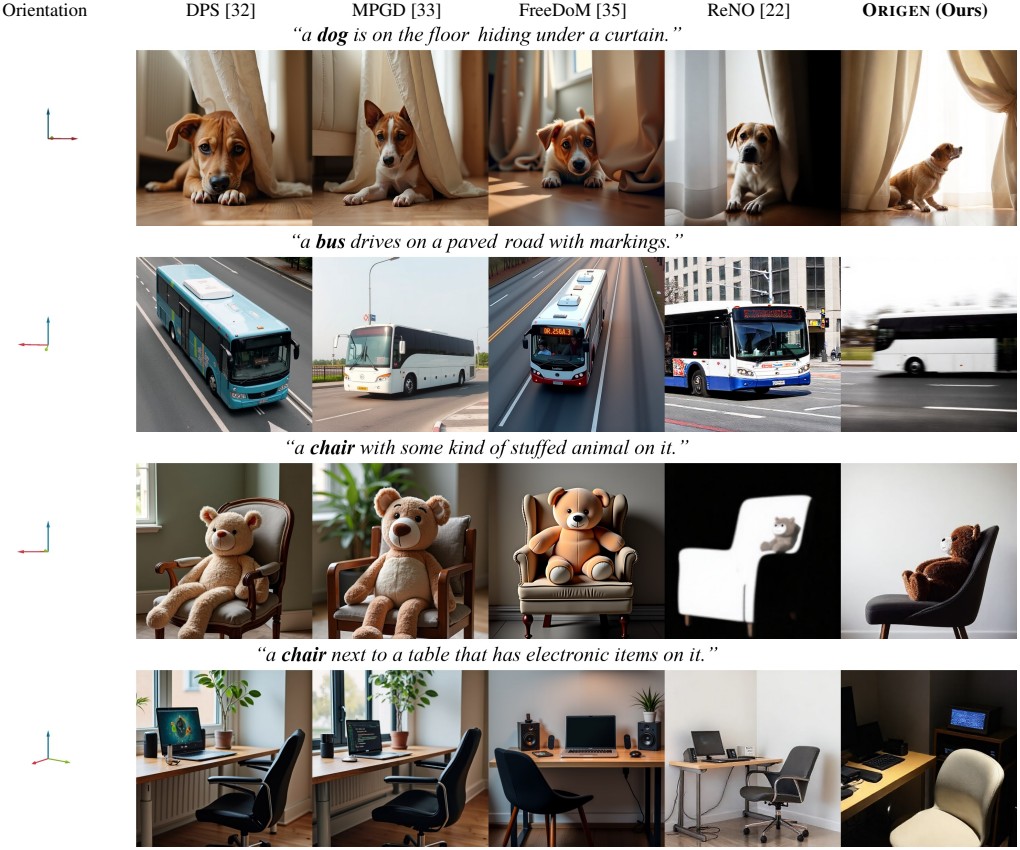

Figure 6: **Qualitative comparisons on generally extended ORIBENCH-Single benchmark.** Compared to other training-free approaches [32, 33, 35, 22], ORIGEN generates the best aligned images with the given orientation grounding conditions.

| Orientation | SD-Turbo [24] | SDXL-Turbo [25] | FLUX-Schnell [21] | Zero-1-to-3 [17] | C3DW [18] | ORIGEN (Ours) |
|---|---|---|---|---|---|---|

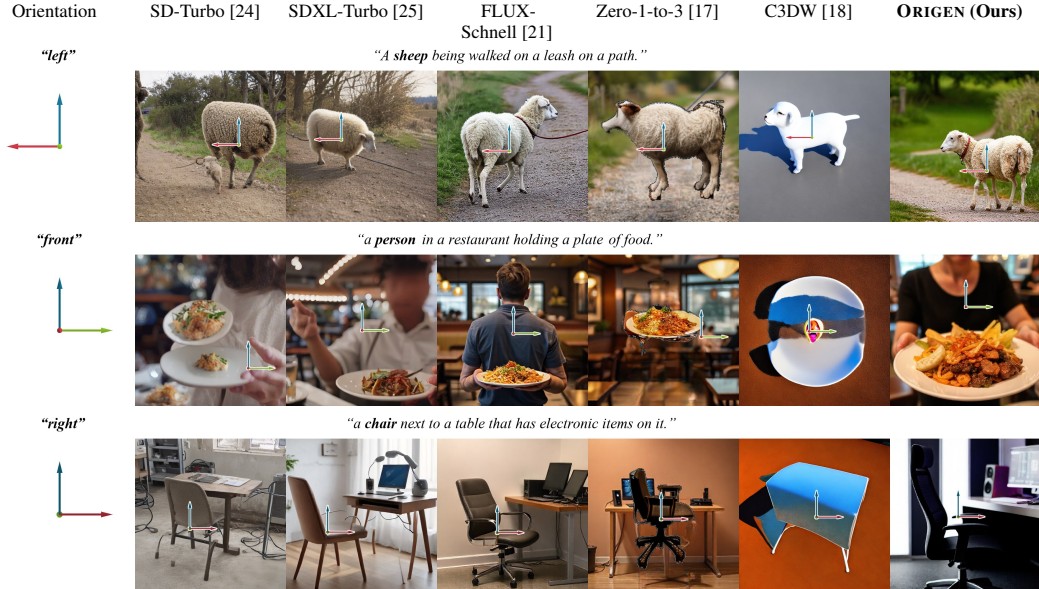

Figure 7: **Qualitative comparisons on extend ORIBENCH-Single benchmark for four primitive orientations.**

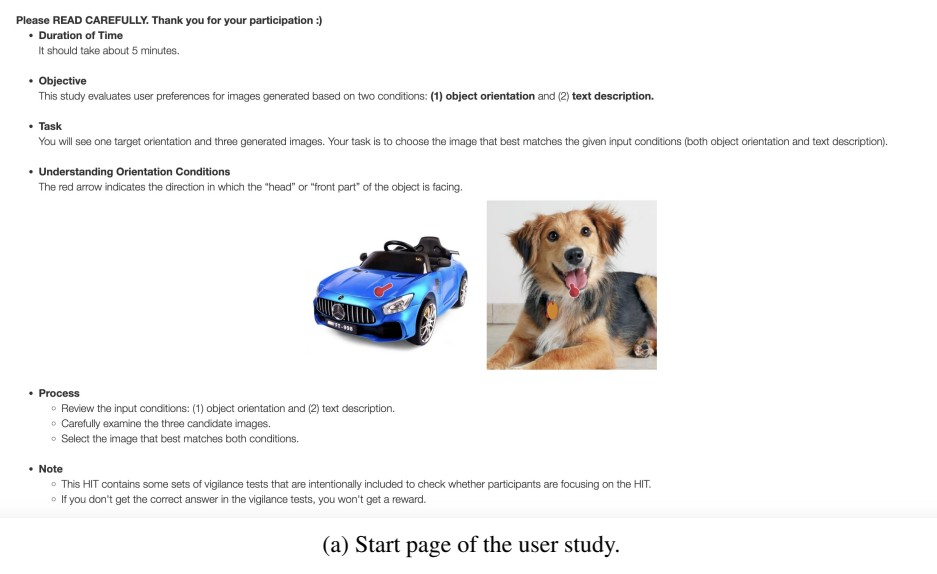

(a) Start page of the user study.

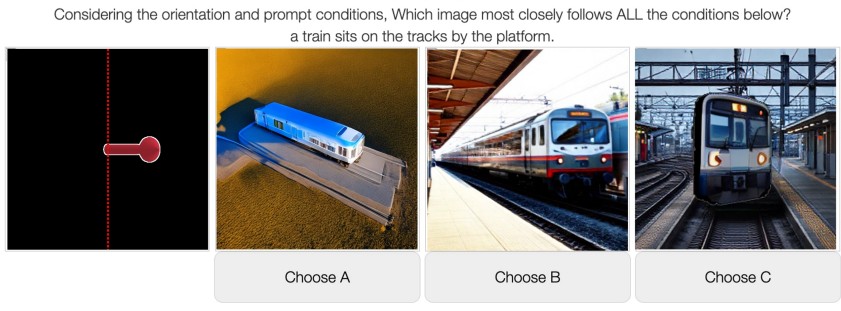

(b) Main test page of the user study.

Figure 8: **Screenshots of our user study.**

at the end of each caption in ORIBENCH-Single dataset. For more comprehensive comparisons, we also included the fine-tuned models (C3DW [18] and Zero-1-to-3 [17]) considered in Sec. 4.5 in this experiment as well. As shown in Tab. 6, ORIGEN significantly outperforms all baseline models in orientation alignment. Note that, although FLUX-Schnell [21] achieves the highest alignment among the vanilla T2I models, ORIGEN surpasses it by more than 2.5 times in the 3-view alignment setting (82.4% vs. 31.2%) and more than 2 times in the 4-view alignment setting (86.6% vs. 42.4%). This substantial margin highlights the inherent ambiguity and lack of precise control in vanilla T2I approaches, as orientation information embedded within textual descriptions is less explicit and reliable compared to direct orientation guidance. We further demonstrate the advantage of ORIGEN through qualitative comparisons in Fig. 7. Vanilla T2I models frequently fail to adhere strictly to the desired orientation, even with directional phrases in text prompts. In contrast, ORIGEN consistently generates images accurately aligned with the specified orientations.

## H  User Study Examples

In this section, we provide details of the user study. To assess user preferences, we conducted an evaluation comparing images generated by Zero-1-to-3 [17], C3DW [18], and ORIGEN using ORIBENCH-Single as the benchmark. The study was conducted on Amazon Mechanical Turk (AMT). For each object class, one sample was randomly selected, resulting in a total of 25 questions.

Table 7: **Computational cost comparison of orientation grounding methods on single NVIDIA A100 GPU.**

| Method | NFEs | Time per Iter (Total) | VRAM | Img Size |
|---|---|---|---|---|
| SD-Turbo [24] | 1 | 0.08s (0.08s) | 4GB | $512 \times 512$ |
| SDXL-Turbo [25] | 1 | 0.30s (0.30s) | 15GB | $1024 \times 1024$ |
| FLUX-Schnell [21] | 1 | 1.01s (1.01s) | 32GB | $512 \times 512$ |
| C3DW [18] | 20 | 6.09s (0.30s) | 6GB | $768 \times 768$ |
| Zero-1-to-3 [17] | 50 | 8.19s (0.16s) | 46GB | $256 \times 256$ |
| DPS [32] | 50 | 53.2s (1.06s) | 43GB | $512 \times 512$ |
| MPGD [33] | 50 | 39.3s (0.79s) | 36GB | $512 \times 512$ |
| FreeDoM [35] | 50 | 52.9s (1.06s) | 43GB | $512 \times 512$ |
| ReNO [22] | 50 | 54.5s (1.09s) | 43GB | $512 \times 512$ |
| **ORIGEN (Ours)** | 50 | 52.7s (1.05s) | 43GB | $512 \times 512$ |

Table 8: **Extended analysis on the reward-adaptive monitor function.** We ablate (i) the slope parameter $k$ and (ii) the step-size bounds $(s_{\min}, s_{\max})$ to examine their effects on the sampling performance. For (i), $(s_{\min}, s_{\max})$ are fixed to their default values of $(\frac{1}{3}, \frac{4}{3})$, and for (ii), $k$ is fixed to its default value of $0.3$. Results indicate that overall performance is robust to variations in these hyperparameters. Best and second-best values are highlighted in **bold** and underlined, respectively.

| (i) Ablation on slope parameter $k$ | | | | | | (ii) Ablation on step-size bounds $(s_{\min}, s_{\max})$ | | | | | | |
|---|---|---|---|---|---|---|---|---|---|---|---|---|
| $k$ | Azi. Acc. ↑ | Azi. Err. ↓ | CLIP ↑ | VQA ↑ | PickScore ↑ | $s_{\min}$ | $s_{\max}$ | Azi. Acc. ↑ | Azi. Err. ↓ | CLIP ↑ | VQA ↑ | PickScore ↑ |
| 0.1 | 0.861 | 17.93 | **0.267** | 0.732 | **0.226** | 0.2 | 1.6 | 0.881 | **16.72** | 0.264 | 0.729 | 0.224 |
| 0.2 | 0.863 | 17.56 | 0.264 | 0.729 | 0.224 | 0.2 | 1.2 | 0.871 | 17.64 | 0.264 | 0.732 | **0.225** |
| 0.3 | 0.871 | 17.41 | 0.265 | **0.735** | 0.224 | 0.333 | 1.333 | 0.871 | 17.41 | 0.265 | **0.735** | 0.224 |
| 0.4 | **0.882** | **17.17** | 0.264 | 0.731 | 0.225 | 0.4 | 1.2 | **0.887** | 16.94 | **0.266** | 0.734 | **0.225** |
| 0.5 | 0.876 | 17.54 | 0.264 | 0.732 | 0.224 | 0.6 | 1.6 | 0.884 | 17.14 | 0.265 | 0.733 | 0.224 |

The orientations used to generate the images were intuitively visualized and presented alongside their corresponding prompts. As shown in Fig. 8, participants were asked to respond to the following question: *"Considering the orientation and prompt conditions, which image most closely follows ALL the conditions below?"*. Each user study session included 25 test samples along with 5 vigilance tests, which were randomly interspersed. The final results were derived from responses of valid participants who correctly answered at least three vigilance tests, leading to a total of 55 valid participants out of 100. No additional eligibility restrictions were imposed.

# I   Computational Costs

In Tab. 7, we compare the computational costs of all baselines and ORIGEN in terms of inference time and GPU memory consumption. Our base model (FLUX-Schnell [21]) and all guided generation methods were evaluated for generating 512×512 images, while other methods followed their default settings. Guided generation methods are generally slower due to the need for reward backpropagation. All methods were evaluated under a setup of 50 Number of Function Evaluations (NFEs). Further analysis demonstrating the fast convergence of our reward-guided Langevin Dynamics is provided in Appendix C.2, suggesting that the computational cost can be partially reduced by using fewer NFEs. All measurements were conducted on a single A100 GPU with 80GB of memory.

# J   Extended Analysis of the Reward-Adaptive Monitor Function

To further understand the effect of our reward-adaptive monitor function introduced in Sec. 3.4, we provide an extended analysis of its hyperparameters and their influence on model performance. As defined in Eq. 9, the monitor function adaptively modulates the sampling step size based on the reward, with three key hyperparameters: (i) $k$ controlling the slope (how fast the step size decreases as the reward increases) and (ii) $(s_{\min}, s_{\max})$ defining the lower and upper bounds of the adaptive step size range.

We systematically vary these hyperparameters to evaluate the sensitivity of our sampling dynamics and its robustness to different configurations. Unless otherwise specified, all other hyperparameters are kept at their default values as reported in the main paper.

As shown in Tab. 8, our performance remains robust across all hyperparameter settings, with only negligible variations in the evaluation metrics. The default values were selected through a coarse grid search rather than fine-tuning, indicating that ORIGEN does not rely on fragile hyperparameter choices. Moreover, all variants with the reward-adaptive monitor further outperform the fixed-step baseline reported in Tab. 1 of the main paper (denoted as ORIGEN*), demonstrating its overall effectiveness.

## K    More Comprehensive Comparisons with Training-Based Methods

To complement the training-based results presented in Tab. 1 of the main paper, we provide more comprehensive quantitative comparisons with existing *training-based* approaches on the ORIBENCH-Single benchmark. Detailed descriptions of the baselines are provided below, and the quantitative results are summarized in the subsequent subsection.

### K.1    Baselines.

**Zero-1-to-3. [17]** Please refer to 'About Zero-1-to-3' section in Appendix F.

**C3DW. [18]** Please refer to 'About C3DW' section in Appendix F.

**DDPO. [77]** We evaluated RL-based fine-tuning method for orientation alignment. Specifically, we adopt DDPO [77], which fine-tunes a diffusion model via policy gradient updates on a learned reward signal. We note that existing RL-based diffusion fine-tuning methods are designed for either unconditional or text-conditional generation [48, 49, 77]. Thus, to enable orientation conditioning while leveraging existing code of DDPO, we opt to appending orientation phrases to the input text prompts (e.g., "object is viewed from front-right, low-angle (azimuth 315°, polar 61°, rotation 90°)"). To fine-tune with our orientation grounding reward, we constructed a single-object text–orientation paired training dataset using 50k orientations from the OrientAnything [20] training dataset and captions from the Cap3D dataset [78, 79]. For other fine-tuning setups (e.g., hyperparameters), we primarily followed the default configuration of original DDPO to ensure fair comparisons.

**SV3D. [80]** SV3D is a multi-view diffusion model generates images conditioned on input image and target view angle. SV3D generates 21 views simultaneously, and we used the one generated image with target azimuth. The whole pipeline used in evaluation is same as Zero-1-to-3 [17], so please refer to 'About Zero-1-to-3' section in Appendix F.

**MV-Adapter. [81]** MV-Adapter is a multi-view diffusion model generates images conditioned on input image and target view angle. MV-Adapter generates 6 views simultaneously, and we used the one generated image with target azimuth. The whole pipeline used in evaluation is same as Zero-1-to-3 [17], so please refer to 'About Zero-1-to-3' section in Appendix F.

### K.2    Results.

As shown in Table 9, ORIGEN achieves substantially higher orientation alignment accuracy than all training-based baselines while maintaining competitive text-to-image alignment performance. These results demonstrate that our inference-time reward-guided sampling consistently outperforms training-based methods that rely on explicit retraining or fine-tuning for orientation control.

## L    Reward-Guided Langevin Dynamics for Other Rewards

To assess the broader applicability of our reward-guided Langevin dynamics sampling, which is inherently reward-agnostic, we extended our experiments beyond 3D orientation grounding to additional tasks, including layout-to-image and depth-guided generation.

Table 9: **Quantitative comparisons on Training-based Methods.** ORIGEN maintains high accuracy even when compared with more recent training-based methods. Best and second-best results are highlighted in **bold** and underlined, respectively.

| Id | Model | Orientation Alignment | | Text-to-Image Alignment | | |
|---|---|---|---|---|---|---|
| | | Azi, Acc. @22.5° ↑ | Azi, Abs. Err. ↓ | CLIP ↑ | VQA ↑ | PickScore ↑ |
| 1 | Zero-1-to-3 [17] | 0.499 | 59.03 | 0.272 | 0.663 | 0.213 |
| 2 | C3DW [18] | 0.426 | 64.77 | 0.220 | 0.439 | 0.197 |
| 3 | DDPO [77] | 0.494 | 40.83 | 0.256 | 0.702 | **0.233** |
| 4 | SV3D [80] | 0.410 | 60.26 | **0.274** | 0.313 | 0.196 |
| 5 | MV-Adapter [81] | 0.486 | 50.19 | 0.204 | 0.313 | 0.196 |
| 6 | **ORIGEN (Ours)** | **0.871** | **17.41** | 0.265 | **0.735** | 0.224 |

Table 10: **Quantitative comparisons on layout-to-image and depth-guided generation.** Best and second-best results are highlighted in **bold** and underlined, respectively.

| Id | Model | Layout-to-Image Generation | | | | | Depth-Guided Generation | | | |
|---|---|---|---|---|---|---|---|---|---|---|
| | | Spatial Alignment | | Text-to-Image Alignment | | | Depth Alignment | Text-to-Image Alignment | | |
| | | mIoU ↑ | HRS ↑ | CLIP ↑ | VQA ↑ | PickScore ↑ | AbsRel ↓ | CLIP ↑ | VQA ↑ | PickScore ↑ |
| 1 | FreeDoM [35] | 0.244 | 60 | 0.261 | 0.640 | 0.261 | 6.34 | 0.256 | 0.672 | 0.223 |
| 2 | ReNO [22] | 0.287 | 70 | 0.261 | 0.655 | **0.229** | 6.67 | **0.265** | **0.705** | **0.233** |
| 3 | **ORIGEN (Ours)** | **0.344** | **72** | **0.284** | **0.659** | **0.229** | **4.62** | 0.264 | 0.686 | 0.230 |

## L.1 Experimental Setup

We used FreeDoM [35] and ReNO [22] as baselines to highlight the effectiveness of our sampling strategy.

**Layout-to-Image Generation.** For evaluation setup, we randomly sampled 100 examples from the HRS-Spatial dataset [82] and used GroundingDINO [54] as the reward model to guide object positioning based on bounding-box conditions. We evaluated the results using mIoU (measured by other detection model [83]) and HRS score.

**Depth-Guided Generation.** For evaluation setup, we randomly sampled 100 image-caption pairs from the ORIBENCH-Single dataset and generated pseudo-ground-truth depth maps using DepthAnything-V2 [84]. We then used DepthAnything-V2 as a reward model to guide image generation and evaluated the results using Absolute Relative Error (AbsRel).

## L.2 Results

As shown in Fig. 9 and Tab. 10, ORIGEN consistently outperforms other guided-generation methods [35, 22] in condition alignment and image quality across both layout-to-image and depth-guided generation tasks. These results demonstrate that our method generalizes effectively to diverse reward alignment tasks.

# M More Qualitative Comparisons on Single Object Orientation Grounding

In the following, Fig. 10 presents additional qualitative comparisons on the single object orientation grounding benchmarks.

# N More Qualitative Comparisons on Multi Object Orientation Grounding

We report more qualitative results on multi object orientation grounding (ORIBENCH-Multi) in Fig. 11.

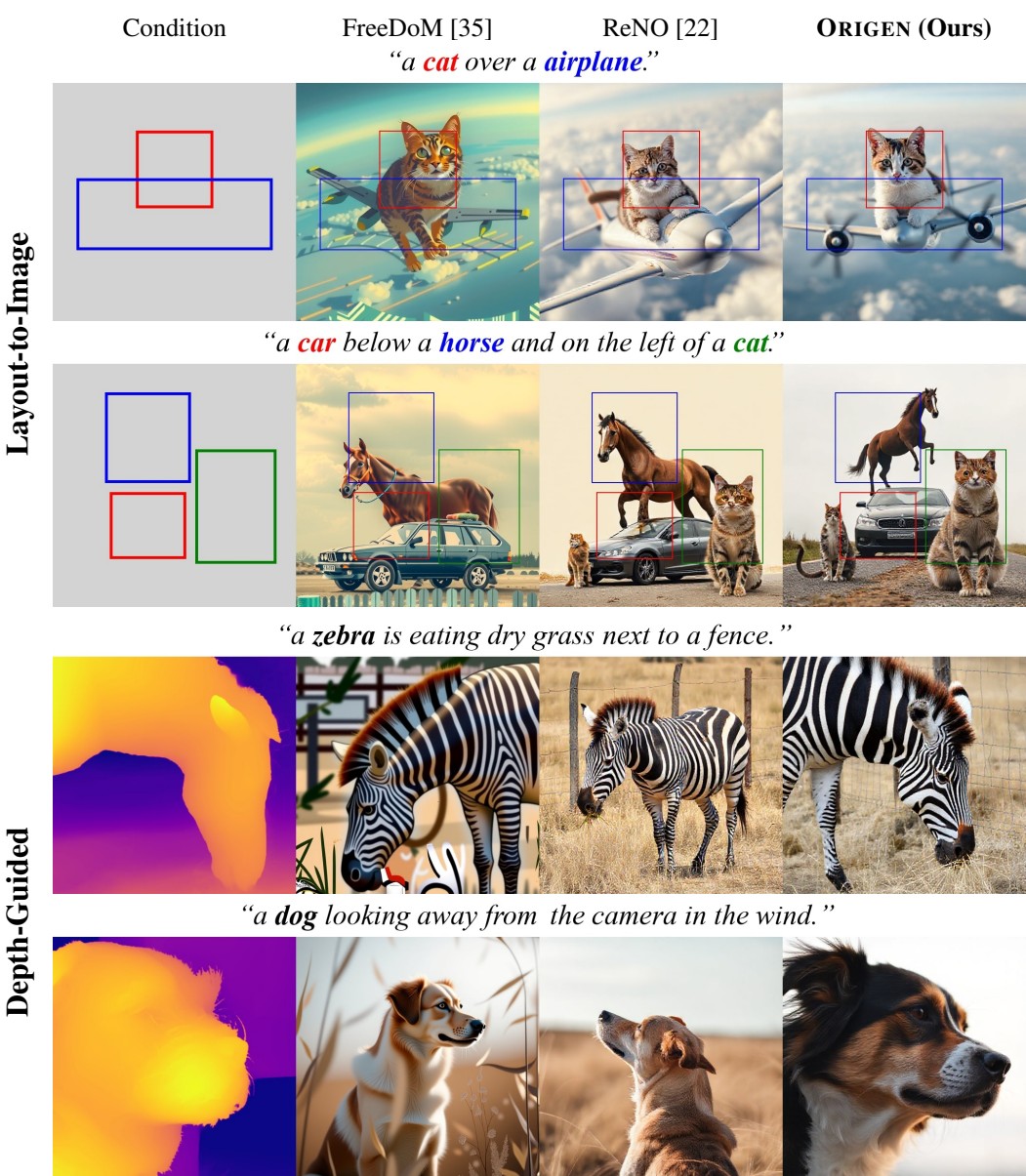

Figure 9: **Qualitative comparisons on layout-to-image (rows 1-2) and depth-guided generation (rows 3-4).** ORIGEN achieves better condition alignment and overall image quality compared to other guided-generation methods [35, 22].

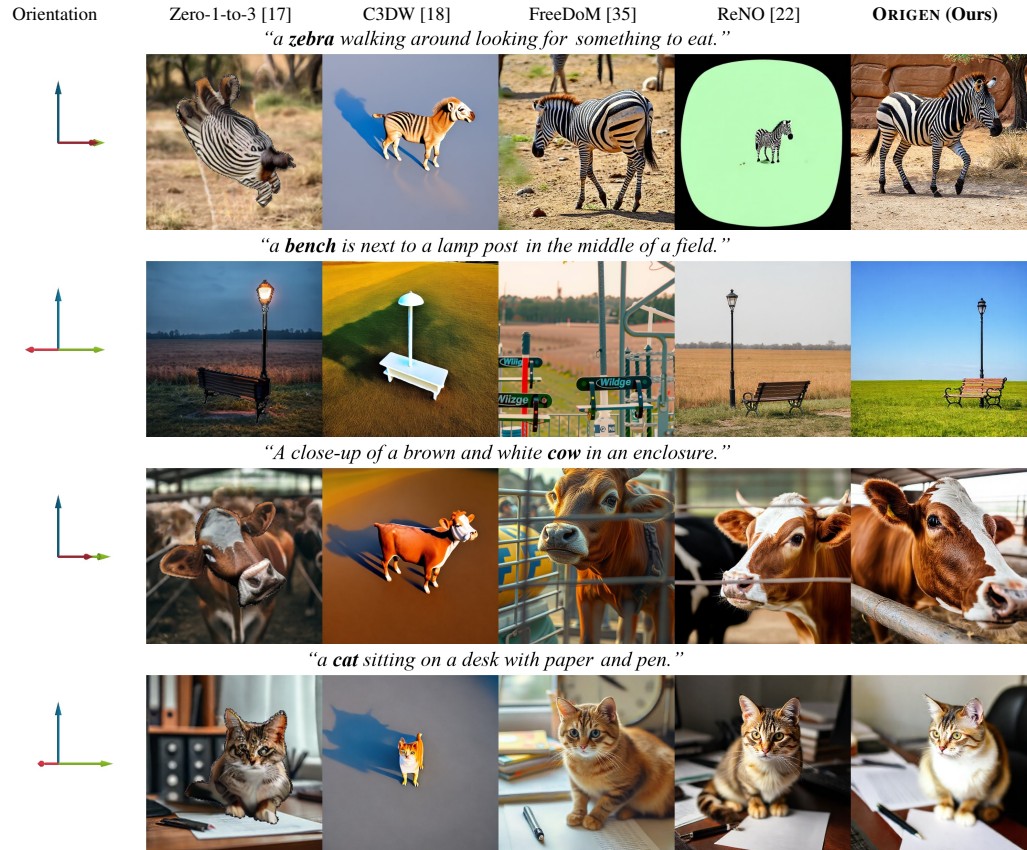

Figure 10: **Qualitative comparisons on ORIBENCH-Single benchmark (Sec. 4.5).** Compared to the existing orientation-to-image models [18, 17], ORIGEN generates the most realistic images, which also best align with the grounding conditions in the leftmost column.

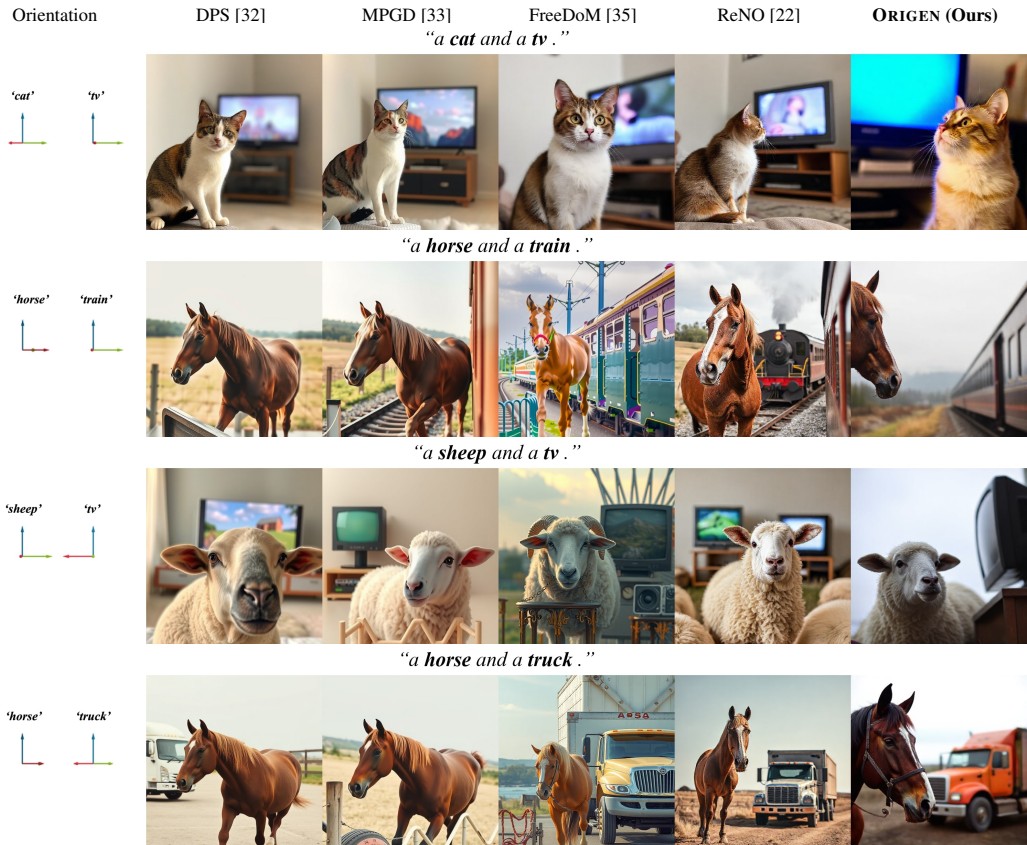

Figure 11: **Qualitative comparisons on ORIBENCH-Multi benchmark (Sec. 4.5).** Compared to the guided-generation methods [32, 33, 35, 22], ORIGEN generates the most realistic images, which also best align with the grounding conditions in the leftmost column.

