# OpenReview forum: "ORIGEN: Zero-Shot 3D Orientation Grounding in Text-to-Image Generation"
_NeurIPS.cc/2025/Conference — NeurIPS 2025 poster_

### Official Review · Reviewer_rW2t · 2025-06-25

**Clarity:** 3
**Significance:** 2
**Originality:** 3
**Rating:** 4
**Confidence:** 3

**Summary:**

ORIGEN introduces the first zero-shot framework that grounds 3-D orientations for multiple objects during text-to-image generation. It combines a one-step latent flow generator with a reward from a pretrained orientation estimator, then performs Langevin-style sampling (gradient + noise) plus reward-adaptive step sizing to maintain realism while enforcing pose. Experiments on a new MS-COCO-based benchmark and user studies show clear gains over training-based baselines (Zero-1-to-3, Continuous3DWords) as well as prompt-only or test-time guidance methods.

**Questions:**

I generally agree with the overall presentation and motivation and therefore lean toward acceptance; however, due to some practical concerns, I currently rate it as borderline accept. I hope the authors will consider the following in their rebuttal:

- Provide more comparative baselines methods.
- Analyze whether the proposed method can be applied to image editing, enabling control over object orientation within images.
- Include additional generated examples to further demonstrate the method’s effectiveness.

**Ethical Concerns:**

["NO or VERY MINOR ethics concerns only"]

**Final Justification:**

The authors have solved most of my questions. Therefore, I have decided to maintain my score as borderline accept.

**Limitations:**

yes

**Paper Formatting Concerns:**

there is no  major formatting issue in this paper.

**Quality:**

3

**Strengths And Weaknesses:**

Strengths

- The visual results presented are highly impressive.
- The paper’s logic is coherent; the toy experiment effectively demonstrates the method’s validity, particularly highlighting the advantage of reward-adaptive time rescaling.
- The experimental design is solid, and the method exhibits clear benefits.

Weaknesses

- The paper’s objective resembles motion control approaches, a direction that has been explored extensively; from my perspective, the results shown here lack some novelty.
- The practical application scope of text-to-image orientation control is rather limited, and I have concerns about the stability of a zero-shot method.
- The most meaningful application seems to be image editing tasks similar to Zero-123 for controlling object orientation, but Zero-123 is already quite old and many multi-view generation methods now exist. The authors should include additional experiments.

---

> ### Author Rebuttal · Authors · 2025-07-30
>
> Thank you for your positive and constructive feedback. We appreciate your recognition of the clarity and coherence of our method, the effectiveness of our reward-adaptive sampling strategy, and the strength of our visual and experimental results. We address your comments in detail below:
>
> **(1) Difference with motion-control approaches.**
>
> As “motion control approaches” may refer to a broad range of domains (e.g., camera trajectory control, object animation), we would appreciate it if the reviewer could kindly clarify which specific line of work is being referenced. We would be glad to address the comparison more directly.
> In general, our method focuses on grounding the static 3D object orientations during image generation from scratch, rather than manipulating their motions over time. As such, we believe our problem formulation and methodology are distinct from works that address temporal motion control.
>
> **(2) Scope and stability of zero-shot approaches.**
>
> **Scope.** Note that spatial grounding in image generation is one of the most active research fields [1, 2, 3, R1, R2]. While the existing spatial grounding methods were limited to 2D position grounding, ORIGEN makes a significant contribution by extending the grounding modality to **3D orientation**, which has only been explored in limited forms in prior works.
>
> **Zero-shot approaches.** We respectfully disagree with the claim that zero-shot methods are generally less stable than training-based methods. As discussed in the paper, training or fine-tuning a generative model conditioned on object orientation is non-trivial due to the lack of a large-scale dataset containing (prompt, image, orientation) samples. Consequently, existing fine-tuning-based approaches such as [18] exhibit clear limitations in both orientation controllability and realism, as evidenced by our comparisons in Tables 1 and 6 in the paper. In contrast, our method does not require training a base model and can fully leverage the rich prior knowledge embedded in large pre-trained text-to-image models. This allows for the generation of high-quality images across diverse prompts, object categories, and orientations, making our approach both practical and effective.
>
> **(3) More comparative baseline methods.**
>
> Following your suggestion, we performed additional baseline comparisons with state-of-the-art multi-view diffusion models more recent than Zero-1-to-3 — SV3D (ECCV 2024) [R3] and MV-Adapter (ICCV 2025) [R4] — on the ORIBENCH-Single benchmark. As shown in the table below, our method outperforms all these additional baselines, demonstrating the effectiveness of the proposed Reward-Guided Langevin Dynamics and Reward-Adaptive Time Rescaling. We thank your suggestion and will include these additional results in the revision.
>
> ---
> **Table.** Comparison on ORIBENCH-Single. SV3D and MV-Adapter generate 6 and 21 views respectively; the one generated image with target azimuth is selected for evaluation. All models use identical prompts and target azimuths. Best results are shown in **bold**, second‑best are *italics*.
> | Method | **Orientation Alignment** |  | **Text‑to‑Image Alignment** |  |  |
> |--------|---------------------------|--|-----------------------------|--|--|
> |        | Azi. Acc. @\\(22.5^{\\circ}\\) ↑ | Azi. Abs. Err. ↓ | CLIP ↑ | VQA ↑ | Pick Score ↑ |
> | Zero-1-to-3           | *0.499* | 59.03 | 0.272 | 0.663 | 0.213 |
> | C3DW           | 0.426 | 64.77 | 0.220 | 0.439 | 0.197 |
> | SV3D [R3] | 0.410 | 60.26 | *0.274* | *0.678* | *0.214* |
> | MV-Adapter [R4] | 0.486 | *50.19* | 0.204 | 0.313 | 0.196 |
> | **ORIGEN (Ours)**   | **0.871** | **17.41** | **0.265** | **0.735** | **0.224** |
>
>
> **(4) Targeted problem scope (vs. image editing).**
>
> While we agree that image editing with object orientation control is a potentially useful application, we did not initially prioritize this task, as spatial grounding itself is considered a stand-alone application in the literature, and the related prior works [1, 2, 3, R1, R2] also do not necessarily address image editing. We believe the nature of these two tasks differs: grounded image generation focuses on generating images from scratch while aligning with grounding conditions, whereas image editing aims to modify an existing image while preserving its original content.
>
> Nevertheless, we agree that image editing with orientation control is another valuable direction. Our method could potentially be extended to this task by integrating personalized generation techniques or incorporating reward terms that encourage content consistency with a reference image [R5, R6]. We promise to include these additional image editing application results in the revision.
>
>
> **(5) Additional generated examples.**
>
> Although we cannot include additional examples due to the NeurIPS policy prohibiting the inclusion of images or videos in the rebuttal, we promise to include additional qualitative results—covering more diverse object categories, captions, and orientations—in the revision. We thank your valuable suggestion.
>
> ---
>
> **References**
>
> **[R1]** BoxDiff: Text-to-Image Synthesis with Training-Free Box-Constrained Diffusion, Xie *et al.,* ICCV 2023\
> **[R2]** R&B: Region and Boundary Aware Zero-shot Grounded Text-to-Image Generation, Xiao *et al.,* ICLR 2024\
> **[R3]** SV3D: Novel Multi-view Synthesis and 3D Generation from a Single Image using Latent Video Diffusion, Voleti *et al.,* ECCV 2024\
> **[R4]** MV-Adapter: Multi-view Consistent Image Generation Made Easy, Huang *et al.,* ICCV 2025\
> **[R5]** Improving Editability in Image Generation with Layer-wise Memory, Kim *et al.,* CVPR 2025 \
> **[R6]** DreamBooth: Fine-Tuning Text-to-Image Diffusion Models for Subject-Driven Generation, Ruiz *et al.,* CVPR 2023

---

> > ### Comment · Reviewer_rW2t · 2025-08-05
> > **Reply to authors**
> >
> > Thank you for your response. Zero-123, MV-Adapter, and SV3D are all trained to assist 3D generation; therefore, I’d like to know whether this method can also aid 3D generation. Of course, if conducting the related experiments is difficult, an analysis alone would be fine.

---

> ### Author Response · Authors · 2025-08-06
>
> Thank you for your comment and for engaging in discussions with us. Indeed, we believe that our method can potentially be extended to support 3D generation. Since _3D generation_ requires orientation-grounded image generation along with content preservation, our orientation grounding method can be combined with a **personalized image generation** technique [R1] to produce multi-view images with consistent content, which can then be used to fit a 3D representation (e.g., NeuS, NeRF) to enable 3D generation. (Here, we assume a 3D generation scheme similar to that of MV-Adapter or SV3D, which reconstructs 3D objects based on the generated multi-view images.)\
> While our method can be extended for 3D generation in this way, we respectfully reiterate that our targeted task—grounded image generation—is considered a different scope in the existing literature [18, 19, 1, 2, 3], as it aims to generate images from scratch (unlike the mentioned baselines, which require a reference image as input) while aligning with grounding conditions. In these prior works [18, 19, 1, 2, 3], grounded image generation has generally been treated as a standalone application, without necessarily addressing additional applications.\
> Nevertheless, we agree that 3D generation can be a valuable application of our method, and we promise to include the corresponding results in the revision.
>
> ---
> **References**\
> **[R1]** DreamBooth: Fine-Tuning Text-to-Image Diffusion Models for Subject-Driven Generation, Ruiz et al., CVPR 2023

---

### Official Review · Reviewer_D7en · 2025-06-28

**Clarity:** 2
**Significance:** 2
**Originality:** 2
**Rating:** 4
**Confidence:** 3

**Summary:**

The paper introduces ​ORIGEN, a zero-shot method for 3D orientation grounding in text-to-image (T2I) generation. Unlike prior work focused on 2D spatial control (e.g., bounding boxes), ORIGEN enables precise 3D orientation control for multiple objects across diverse categories without requiring task-specific training data. The key innovation is a ​reward-guided sampling approach​ leveraging a pretrained discriminative model (OrientAnything) for orientation estimation and a one-step generative flow model (FLUX-Schnell). To address challenges like local optima and realism degradation in gradient-based optimization, ORIGEN proposes ​Langevin dynamics-based sampling​ with noise injection and ​reward-adaptive time rescaling​ for faster convergence. Experiments on the curated ORIBENCH benchmark (MS-COCO-based) show ORIGEN outperforms training-based (e.g., Zero-1-to-3, C3DW) and test-time guidance baselines (e.g., DPS, ReNO) in orientation alignment while maintaining text-image fidelity. User studies further validate its superiority (58.18% preference).

**Questions:**

1. ​Failure Cases: What are the most common failure modes (e.g., misalignment due to OrientAnything’s errors)? Would combining text prompts with explicit orientation cues improve robustness?
2. ​Reward Function Flexibility: Can the sampling method be adapted to different reward functions for broader controllability? If yes, how does it compare to other guidance-based approaches?

**Ethical Concerns:**

["NO or VERY MINOR ethics concerns only"]

**Final Justification:**

The authors have provided additional quantitative results on diverse 3D control, spatial grounding, and depth-conditioned generation, which address some of my concerns. Given these improvements, this method could serve as a promising training-free alternative for guided image generation. Accordingly, I have decided to raise my score to ​Borderline Accept—provided the authors include these additional experiments in the final version.

**Limitations:**

yes

**Quality:**

2

**Strengths And Weaknesses:**

Strengths

1. ​Novelty: First zero-shot method for multi-object 3D orientation grounding in T2I generation, addressing a gap in prior work limited to single-object/synthetic data.
2. ​Strong Empirical Results: Outperforms baselines on ORIBENCH (85.4% Acc.@22.5° for single-object, 69.2% for multi-object) and user studies, demonstrating generalization to real-world images.


Weaknesses​

1. ​Limited 3D Control: While object orientation is represented by three Euler angles, the results primarily demonstrate control over the xy-plane. The lack of variation in the upright (z-axis) direction suggests the method may not fully exploit 3D orientation grounding.
​2. Missing Ablation Studies: Given that ORIGEN is training-free, conducting ablation studies (e.g., validating the impact of ​Reward-Guided Langevin Dynamics​ and ​Reward-Adaptive Time Rescaling) would strengthen the claims. Such experiments should be computationally feasible and could provide deeper insights into the method’s components.
3. ​Generalizability to Other Controls: The proposed sampling method is agnostic to the reward function. Could it be extended to other spatial attributes (e.g., pose, depth) by swapping the reward function? If so, a comparison with training-free guided methods like ​FreeDoM​ would highlight its versatility.

---

> ### Author Rebuttal · Authors · 2025-07-30
>
> Thank you for your thoughtful and encouraging review. We appreciate your recognition of our method’s novelty and strong empirical results. Below, we provide detailed responses to your comments:
>
> **(1) Limited 3D control.**
>
> While our model is fully capable of controlling the complete 3D orientation—including variations along the z-axis—we designed our experimental setup within the xy-plane to enable comparisons with the baseline methods, which do not support full 3D controllability (L244-246). While we are unable to show additional image results here due to the NeurIPS policy prohibiting the inclusion of images in the rebuttal, we promise to add more such examples in the revised version to better showcase the versatility of our approach beyond the xy-plane.
>
> **(2) Ablation study on Langevin Dynamics.**
>
> Please note that we have already included experiments validating both our Reward-Guided Langevin Dynamics and Reward-Adaptive Time Rescaling in Table 1 in the paper. To evaluate the effectiveness of Reward-Guided Langevin Dynamics, we included ReNO [22], which uses gradient ascent with noise regularization, as a baseline in the table.
>
> For a more comprehensive comparison, here, we show additional comparisons with a naïve gradient ascent method (i.e., ReNO without noise regularization) in the table below. Our method achieves superior results compared to all these baselines, demonstrating the advantages of Reward-Guided Langevin Dynamics in both orientation alignment and text-to-image alignment.
>
> To validate the effectiveness of Reward-Adaptive Time Rescaling, we already provided an ablation in Table 1 in the main paper, comparing ORIGEN* (without time rescaling) and ORIGEN (with time rescaling), where our final method with time rescaling achieves clearly improved performance.
>
>
> | Method           | **Orientation Alignment** |   | **Text‑to‑Image Alignment** |    |    |
> |------------------|----------------------------------|-------------------------|------------|-|---|
> |                  | Azi. Acc. @ 22.5 ↑ | Azi. Abs. Err. ↓        | CLIP ↑                      | VQA ↑  | PickScore ↑  |
> | Naïve GA         | 0.745    | 21.47     | 0.246          | 0.660 | 0.210   |
> | ReNO [22]        | *0.796*   | *20.56*         | *0.247*   | *0.663* | *0.212* |
> | **ORIGEN (Ours)**| **0.871**            | **17.41**  | **0.265**    | **0.735** | **0.224** |
>
>
> *Best results are shown in **bold**, second‑best are *italics*. *
>
>
> **(3) Other reward function experiments.**
>
> Thank you for your thoughtful suggestion regarding the potential extensions of our work. As you kindly pointed out, our method is reward-agnostic and can be potentially extended to other reward alignment tasks such as spatial grounding, depth, or pose. To demonstrate its generality, we conducted additional experiments on **spatial grounding** (i.e., bounding-box-conditioned 2D position control) and **depth-conditioned image generation**. In both setups, we used FreeDoM [35] and ReNO [22] as baselines to highlight the effectiveness of our sampling strategy.
>
> ---
>
> **Spatial Grounding.** For this experiment, we randomly sampled 100 examples from the HRS-Bench dataset [R1]. We used GroundingDINO [55] as the reward model to guide object positioning based on bounding-box conditions. As evaluation metrics, we report spatial accuracy using mIoU (measured by SalienceDETR [R2]) and HRS score, and text-image alignment using CLIP similarity, VQA score, and PickScore.
>
> | Method            | **Spatial Alignment** |              | **Text-to-Image Alignment** |        |               |
> |------------------|------------------------|--------------|-----------------------------|--------|---------------|
> |                  | mIoU ↑      | HRS ↑        | CLIP ↑         | VQA ↑  | PickScore ↑   |
> | FreeDoM [35]      | 0.244                  | 60        | 0.261                       | 0.640  | *0.221*          |
> | ReNO [22]         | *0.287*                  | *70*        | *0.283*                     | *0.655*  | **0.229**        |
> | **ORIGEN (Ours)**| **0.344**              | **72**    | **0.284**                   | **0.659** | **0.229**  |
>
> ---
>
> **Depth-Conditioned Generation.** For this setup, we randomly sampled 100 image-caption pairs from the ORIBENCH-Single dataset and generated pseudo-ground-truth depth maps using DepthAnything-V2 [R3]. We then used DepthAnything-V2 as a reward model to guide image generation and evaluated the results using Absolute Relative Error (AbsRel).
>
> | Method            | **Depth Alignment**  | **Text-to-Image Alignment** |        |               |
> |------------------|----------------------|-----------------------------|--------|---------------|
> |                  | AbsRel ↓            | CLIP ↑        | VQA ↑                       | PickScore ↑ |
> | FreeDoM [35]      | *6.34*               | 0.256         | 0.672                       | 0.223        |
> | ReNO [22]         | 6.67              | **0.265**       | **0.705**                       | **0.233**      |
> | **ORIGEN (Ours)**| **4.62**           | *0.264*     | *0.686*                   | *0.230*    |
>
> These results demonstrate that our method is reward-agnostic and effectively generalizes to different types of guidance signals, such as 2D spatial constraints and depth cues, without requiring task-specific model modifications. While this was originally raised as a limitation of our method, *we believe it instead highlights our method’s strength—its generalizability and potential applicability to broader tasks in the future*.
>
>
> **(4) Discussion on failure cases and text prompts with explicit orientation cues.**
>
> **Failure cases.** Generating images of objects with inherently ambiguous orientations (e.g., objects with little distinction between front and back) can indeed be challenging to handle in our framework. We will make sure to clarify these limitations in the revised version of the paper.
>
> **Text prompts with explicit orientation cues.** We conducted additional ablation experiments using text prompts augmented with explicit orientation cues. We considered two settings: (1) adding text prefixes with abstracted orientations (e.g., “{object} is viewed from front-right”), and (2) adding text prefixes with both abstracted and numerical orientations (e.g., “{object} is viewed from front-right, low-angle (azimuth 315°, polar 61°, rotation 90°)”). As shown in the table below, these settings result in negligible effects in orientation grounding performance on the ORIBENCH-Single benchmark, suggesting that orientation grounding cannot be trivially improved by simply adding explicit orientation phrases to the prompt—reinforcing the necessity of our reward-guided sampling approach. We will include this discussion in the revised version of the paper.
>
> | Method             | **Azi. Acc. @ 22.5° ↑** | **Azi. Abs. Err. ↓** |
> |--------------------|--------------------------|------------------------|
> | Abstract        | *0.867*                   | *17.96*                 |
> | Abstract + Numerical           | 0.856                    | 18.51                 |
> | **ORIGEN (Ours)**  | **0.871**                | **17.41**             |
>
> ---
>
> **References**
>
> **[R1]** HRS-Bench: Holistic, Reliable and Scalable Benchmark for Text-to-Image Models, Bakr *et al.,* ICCV 2023\
> **[R2]** Salience DETR: Enhancing Detection Transformer with Hierarchical Salience Filtering Refinement, Hou *et al.,* CVPR 2024\
> **[R3]** Depth Anything V2, Yang *et al.,* NeurIPS 2024

---

> > ### Comment · Reviewer_D7en · 2025-08-06
> >
> > I thank the authors for their detailed responses. Unfortunately, NeurIPS does not permit additional images in the rebuttal, so I cannot evaluate further results on diverse 3D control, spatial grounding, or depth-conditioned generation. That said, the authors have provided additional quantitative results that address some of my concerns. As such, I have decided to raise my score to ​Borderline Accept—on the condition that the authors include these additional experiments in the final version.

---

### Official Review · Reviewer_6FJV · 2025-06-30

**Clarity:** 3
**Significance:** 2
**Originality:** 2
**Rating:** 3
**Confidence:** 4

**Summary:**

This paper introduces ORIGEN, a zero-shot method for 3D orientation grounding in T2I generation. It proposes a reward-guided sampling method based on Langevin dynamics using a one-step generative model and a pretrained orientation estimator. It also designs a novel time-rescaling strategy to accelerate convergence. Experimental results on a new benchmark (ORIBENCH) show significant improvements over prior methods in both single- and multi-object scenarios.

**Questions:**

Please respond to the questions in the Weakness section.

**Ethical Concerns:**

["NO or VERY MINOR ethics concerns only"]

**Final Justification:**

After carefully reading the reviews and the authors' response, I agree that the techniques in the submission are, to some extent, valuable for the community. However, the performance is heavily reliant on a single pretrained model for orientation estimation. From my perspective, this is significantly different from the LLM references claimed in the authors' rebuttal due to two main reasons. The first one is that the orientation estimation is a super-narrow research sub-area. This is different from general LLMs, where many foundational models are available. Therefore, relying on a single model makes it significantly vulnerable. Secondly, the pretrained model plays a vital role in this work, and the authors do not sufficiently show the generalizable ability of the proposed techniques.

After comprehensively considering these aspects, I decided to raise my rating to Borderline Reject.

**Limitations:**

yes

**Quality:**

2

**Strengths And Weaknesses:**

Strengths:
1.First zero-shot method for 3D orientation grounding across multiple objects/diverse categories, addressing a critical gap in controllable generation.
2.The proposed Langevin-based sampling approach is elegant, theoretically grounded, and simple to implement. It balances reward maximization and latent prior adherence.
3.The paper is well-written, with clear problem formulation and method descriptions.
4.The proposed reward function does not depend on a specific model architecture or dataset, which improves generality and future extensibility to other basic tasks (e.g., pose, layout).

Weaknesses:
1.The method is heavily reliant on Orient-Anything, a pretrained orientation estimator—its limitations directly affect ORIGEN’s performance and generalization.
2.The method, while elegant, still requires ~50 iterations (as much as multi-step methods), making the “one-step model” narrative less compelling in practice.
3.While the paper provides the values for hyperparameters used in reward-adaptive monitor function, it lacks an in-depth discussion on their sensitivity or the methodology for their selection, merely stating that these configurations provide a 'favorable balance'.
4.In Section 3.1 (Why Test-Time Guidance?), the authors argue that reinforcement learning–based fine-tuning leads to image quality degradation. However, this assertion is not supported by any quantitative or qualitative evidence in the paper. No RL-based simplified baseline is included in the experimental comparisons.
5.In Section 3.1 (Why Based on a One-Step Generative Model?), the authors attribute the poor performance of reward-guided optimization in diffusion models to intermediate-step issues (e.g., blurry early-stage outputs and minimal updates in late stages). However, the paper lacks concrete analysis (e.g., reward gradient magnitude over timesteps) or direct comparison with modern alternatives such as initial-noise optimization or terminal-sample guidance (e.g., InitNO, D-Flow). The dismissal of multi-step diffusion approaches appears oversimplified without sufficient experimental support.

---

> ### Author Rebuttal · Authors · 2025-07-30
>
> We sincerely appreciate your recognition of our work as the first zero-shot framework for 3D orientation grounding. Below, we respond to your concerns in detail:
>
> **(1) Reliance on the *OrientAnything* reward model.**
>
> Although the neural reward model (OrientAnything [20]) might not be perfect and could potentially affect our framework’s performance, our experimental results nevertheless demonstrate that our approach *significantly outperforms* existing orientation grounding methods (Table 1 in the paper).
>
> We also note that, in related works on reward alignment for diffusion models [R1, R2, R3], most methods rely on neural reward models [R4, R5, R6] to enable alignment with objectives that are difficult to analytically define (e.g., text-to-image alignment, aesthetic preference). A similar trend is also widely observed in LLM reasoning, where neural reward models are used to evaluate and guide multi-step reasoning processes [R7, R8]. We similarly follow this paradigm to use a pretrained neural model to define a reward function tailored to our task.
>
> **(2) Sampling ~50 steps, making the one-step model narrative less compelling.**
>
> As discussed in Sec. 3.1 (Why Based on a One-Step Generative Model?), our main purpose for using a one-step generative model is to enable clearer and more reliable reward estimation, rather than to speed up image generation. Note that our reward model (OrientAnything [20]) was trained on, and expects, a clear image as input to compute accurate rewards. However, images generated by multi-step generative models in early steps are often blurry due to the high variance in their marginal distribution. For further discussion on this particular point, we kindly refer the reader to (5) below.
>
> Although efficiency was not our primary motivation for adopting a one-step generative model, we also provided a computation time analysis in Sec. 4.7 in the paper. There, we discussed that our method’s number of function evaluations (NFEs) was set to 50 to perform fair comparisons with the multi-step baselines to match the number of their denoising steps. In practice, our method requires only an average of 12.8 NFEs—*approximately 4× fewer than the 50 NFEs used in our reported experiments*—to achieve superior orientation alignment than all the baseline methods. Please refer to Sec. 4.7 for the full analysis.
>
> **(3) In-depth discussion about Reward-adaptive Monitor Function.**
>
> In our proposed monitor function, \\(\\mathcal{G}(x) = s_{\\min} - \\tanh (k x) \\cdot (s_{\\max} - s_{\\min})\\), hyper-parameters include \\(k \\) controlling the *slope* (how the step size shrinks as the reward rises), and \\(s_{\\min}, s_{\\max} \\) bounding the step size values. Below, we present additional experimental results to assess the sensitivity of these hyper-parameters; note that hyper-parameters are set to their default values in the main paper unless otherwise specified.
>
> #### (i) Ablation on \\(k \\)
>
> | \\(k\\) | **Orientation Alignment** |  | **Text‑to‑Image Alignment** |  |  |
> |--------|---------------------------|--|-----------------------------|--|--|
> |     |  Azi. Acc. @\\(22.5^{\\circ}\\) ↑ | Azi. Abs. Err. ↓ | CLIP ↑ | VQA ↑ | Pick Score ↑ |
> |   0.1 | 0.861 | 17.93 | **0.267** | *0.732* | **0.226** |
> |   0.2 | 0.863 | 17.56 | 0.264 | 0.729 | 0.224 |
> |   0.3 | 0.871 | *17.41* | *0.265* | **0.735** | 0.224 |
> |  0.4 | **0.882** | **17.17** | 0.264 | 0.731 | *0.225* |
> |  0.5 | *0.876* | 17.54 | 0.264 | *0.732* | 0.224 |
>
> #### (ii) Ablation on \\(s_{\\min}\\), \\(s_{\\max}\\)
>
> | \\(s_{\\min}\\) | \\(s_{\\max}\\) | **Orientation Alignment** |  | **Text‑to‑Image Alignment** |  |  |
> |--|--|-------------------------|--|-----------------------------------|-|-|
> |     |     |  Azi. Acc. @\\(22.5^{\\circ}\\) ↑ | Azi. Abs. Err. ↓ | CLIP ↑ | VQA ↑ | Pick Score ↑ |
> | 0.2 | 1.6 | 0.881 | **16.72** | 0.264 | 0.729 | *0.224* |
> | 0.2 | 1.2 | 0.871 | 17.64 | 0.264 | 0.732 | **0.225** |
> | 0.333 | 1.333 | 0.871 | 17.41 | *0.265* | **0.735** | *0.224* |
> | 0.2 | 1.2 | **0.887** | *16.94* | **0.266** | *0.734* | **0.225** |
> | 0.6 | 1.6 | *0.884* | 17.14 | *0.265* | 0.733 | *0.224* |
>
> *Best results are shown in **bold**, second‑best are *italics*. *
>
> **Take-away.** Our performance is *robust* to all three hyper-parameters: changing \\(k, s_{\\min}, s_{\\max} \\) has negligible impact on all evaluation metrics. Our defaults were chosen by a coarse grid search, not to finely tune peak scores, confirming that ORIGEN does not rely on fragile hyper-parameter selection. Moreover, all variants with the reward-adaptive monitor consistently outperform the setting without it (Table 1 in main paper), highlighting its overall effectiveness.
>
>
> **(4) Comparison with RL-based fine-tuning baseline.**
>
> Following your suggestion, we provide additional comparison results with a well-known RL-based fine-tuning method, DDPO [50], which fine-tunes a diffusion model via policy gradient updates on a learned reward signal. We note that existing RL-based diffusion fine-tuning methods are designed for either unconditional or text-conditional generation [48, 50, 51]. Thus, to enable orientation conditioning while leveraging existing code of DDPO, we opt to appending orientation phrases to the input text prompts (e.g., “~. {object} is viewed from front-right, low-angle (azimuth 315°, polar 61°, rotation 90°)”).
> To fine-tune with our orientation grounding reward, we constructed a single-object text–orientation paired training dataset using 50k orientations from the OrientAnything training set and captions from the Cap3D dataset [R9, R10]. For other fine-tuning setups (e.g., hyper-parameters), we primarily followed the original DDPO configuration to ensure fair comparisons, and we will release our full implementation.
>
> As shown in the table below, DDPO-based fine-tuning demonstrates limited improvement of the orientation alignment over the base model (FLUX.1-dev). Moreover, it also results in substantial degradation in image quality, as we described in L106–109. In contrast, our method achieves the highest performance in both orientation and text-to-image alignment, while preserving image fidelity.
>
> | Method | **Orientation Alignment** |  | **Text‑to‑Image Alignment** |  |  |
> |--------|---------------------------|--|-----------------------------|--|--|
> |        | Azi. Acc. @\\(22.5^{\\circ}\\) ↑ | Azi. Abs. Err. ↓ | CLIP ↑ | VQA ↑ | Pick Score ↑ |
> | Base Model (FLUX.1-dev) [21] |  0.395 | 52.14 | **0.268** | **0.739** | **0.234** |
> | DDPO [50] | *0.494* | *40.83* | 0.256 | 0.702 | 0.223 |
> | **ORIGEN (Ours)**   | **0.871** | **17.41** | *0.265* | *0.735* | *0.224* |
>
>
> **(5) Lack of analysis about multi-step guided generation methods**
>
> Please note that we already included comparisons with multi-step diffusion-based methods, which are DPS, MPGD, FreeDoM in Tables 1 and 5 of the paper, where we experimentally validated their suboptimal performance. We respectfully clarify that the phenomenon of "blurry early-stage outputs and minimal updates in late stages" is widely discussed in the existing literature [R11, R12, R13, R14]. For visual evidence, please refer to Fig. 5 and Fig. 6 in DDPM [R15], and Fig. 1 in EDM [R16], which illustrate the blurry Tweedie estimations in the early timesteps.
>
> For the additional baselines suggested in the review (InitNO, D-Flow), both approaches are either infeasible (InitNO) or lack accessible implementations (D-Flow):
>
> **InitNO [45]** was originally designed for an attention score reward, which can be computed after very few reverse steps (typically 1-step). However, the orientation grounding reward in our task must be evaluated on the clean image, requiring backpropagation through much larger timesteps—making it infeasible given current GPU memory limitations.
>
> **D-Flow [43]** addresses the memory issue via heavy gradient checkpointing, but no public code is available. We contacted the authors to request the implementation but did not receive a response. Aside from this code accessibility issue, we note that D-Flow reports a computation time of *3–40 minutes* per prompt, making it orders of magnitude slower than our method.
>
>
> ---
>
> **References**
>
> **[R1]** Diffusion Model Alignment Using Direct Preference Optimization, Wallace *et al.,* CVPR 2024\
> **[R2]** Test-time Alignment of Diffusion Models without Reward Over-optimization, Kim *et al.,* ICLR 2025\
> **[R3]** Training Diffusion Models with Reinforcement Learning, Black *et al.,* ICLR 2024\
> **[R4]** Laion Aesthetic Predictor, Schuhmann *et al.,* 2022\
> **[R5]** ImageReward: Learning and Evaluating Human Preferences for Text-to-Image Generation, Xu *et al.,* NeurIPS 2023\
> **[R6]** Pick-a-Pic: An Open Dataset of User Preferences for Text-to-Image Generation, Kirstain *et al.,* NeurIPS 2023\
> **[R7]** Training verifiers to solve math word problems, Cobbe *et al.,* arXiv 2110.14168\
> **[R8]** Let's Verify Step by Step, Lightman *et al.,* ICLR 2024\
> **[R9]** Objaverse: A Universe of Annotated 3D Objects, Deitke *et al.,* CVPR 2023\
> **[R10]** Scalable 3D Captioning with Pretrained Models, Luo *et al.,* NeurIPS 2023\
> **[R11]** eDiff-I: Text-to-Image Diffusion Models with an Ensemble of Expert Denoisers, Balaji *et al.,* arXiv 2211.01324\
> **[R12]** Prompt-to-Prompt Image Editing with Cross Attention Control, Hertz *et al.,* ICLR 2023\
> **[R13]** Perception Prioritized Training of Diffusion Models, Choi *et al.,* CVPR 2022\
> **[R14]** R&B: Region and Boundary Aware Zero-shot Grounded Text-to-image Generation, Xiao *et al.,* ICLR 2024\
> **[R15]** Denoising Diffusion Probabilistic Models, Ho *et al.,* NeurIPS 2020\
> **[R16]** Elucidating the Design Space of Diffusion-Based Generative Models, Karras *et al.,* NeurIPS 2022

---

> > ### Comment · Reviewer_6FJV · 2025-08-05
> >
> > Thank you for your detailed response and further explanations.

---

> > > ### Author Response · Authors · 2025-08-06
> > >
> > > Thank you for reviewing our rebuttal. We would like to kindly ask whether it has sufficiently addressed your concerns. If there are any remaining points you would like us to clarify, we would appreciate your further feedback and would be glad to discuss them.

---

### Official Review · Reviewer_SEJH · 2025-07-03

**Clarity:** 3
**Significance:** 3
**Originality:** 3
**Rating:** 5
**Confidence:** 3

**Summary:**

This paper introduces a zero-shot method for 3D orientation-conditioned text-to-image generation across multiple object categories. It relies on an off-the-shelf 3D orientation estimation model and a one-step text-to-image generative model. The technical contributions include:
1. A sampling-based reward maximization approach using Langevin dynamics: inject random noise and perform gradient ascent.
2. An adaptive time rescaling method based on the reward function (smaller steps in high-reward regions and larger steps in low-reward regions).

Experiments include:
- A new benchmark comparing the proposed method with SOTA methods in terms of both orientation and text-to-image alignment.
- A user study comparing the proposed method with other SOTA text-to-image methods.

**Questions:**

See weaknesses.

**Ethical Concerns:**

["NO or VERY MINOR ethics concerns only"]

**Final Justification:**

The rebuttal addresses the issue raised by the reviewer. The final recommendation is Accept.

**Limitations:**

yes

**Quality:**

3

**Strengths And Weaknesses:**

**Strengths:**

1. The zero-shot method for 3D orientation-conditioned text-to-image generation is novel and interesting.
2. The sampling-based reward maximization approach and adaptive time rescaling method are technically sound.
3. The proposed benchmark contributes to the community.
4. The presentation quality is good; the figures are clear and the writing is easy to follow.
5. The toy experiment presented in Fig. 2 is a nice addition.

**Weaknesses:**

1. There is no ablation study for the sampling-based reward maximization approach. It would be better to conduct an apple-to-apple comparison with a naive gradient ascent baseline.
2. A video visualization of continuous orientation change (e.g., the object rotating along an axis) would help demonstrate whether the object can truly follow the intended axis.

---

> ### Author Rebuttal · Authors · 2025-07-30
>
> Thank you for your thoughtful and encouraging review. We sincerely appreciate your recognition of the novelty and technical soundness of our approach. We’re also grateful for your positive feedback on our reward-guided sampling, benchmark contributions, and the clarity of our presentation. Below, we provide detailed responses to your comments:
>
> **(1) Comparison with naïve gradient‑ascent method.**
>
> As a kind reminder, we note that we already included comparisons with ReNO [22] (Tables 1 and 5), which is the naïve gradient ascent with noise regularization. The suggested comparison setting—the naïve gradient ascent method—can be interpreted as a simplified version of ReNO without noise regularization, and was not initially prioritized in our experiments, as we considered it an ablation on different types of regularization of ReNO.
>
> Nevertheless, following your suggestion, we conducted an additional comparison with this naïve gradient ascent baseline. As shown in the table below, this method (denoted by Naïve GA) performs the worst in both orientation accuracy and text-to-image alignment. These results further highlight that the stochastic exploration inherent in our Langevin dynamics not only mitigates deviation from the original distribution but also improves alignment with the target reward signal. We thank your comment and will include these results in the revision.
>
> | Method | **Orientation Alignment** |  | **Text‑to‑Image Alignment** |  |  |
> |--------|---------------------------|--|-----------------------------|--|--|
> |        | Azi. Acc. @\\(22.5^{\\circ}\\) ↑ | Azi. Abs. Err. ↓ | CLIP ↑ | VQA ↑ | Pick Score ↑ |
> | Naïve GA            | 0.745 | 21.47 | 0.246 | 0.660 | 0.210 |
> | ReNO [22]           | *0.796* | *20.56* | *0.247* | *0.663* | *0.212* |
> | **ORIGEN (Ours)**   | **0.871** | **17.41** | **0.265** | **0.735** | **0.224** |
>
> *Best results are shown in **bold**, second‑best are *italics*. *
>
> **(2) Video visualization of continuous orientation change.**
>
> Thank you for your helpful suggestion. We agree that adding a video visualization of continuous orientation change would be informative. Although we cannot directly show our video results due to the NeurIPS rebuttal policy that prohibits the inclusion of images or videos, we promise to add these video visualizations in the revision.

---

> > ### Comment · Reviewer_SEJH · 2025-08-04
> >
> > Thanks for adding the extra comparison with the naive gradient ascent method. It helps support your claim. I will keep my original score.

---

> > > ### Author Response · Authors · 2025-08-06
> > >
> > > Thank you for your response and for taking the time to review our additional comparison. We appreciate your thoughtful evaluation.

---

### Note · Authors · 2025-08-13

We sincerely thank the AC and reviewers for their time, effort, and insightful feedback, which has been invaluable in improving the quality of our paper.

Following the rebuttal, three reviewers (D7en, SEJH, rW2t) explicitly confirmed that their concerns had been addressed and recommended acceptance. Unfortunately, we did not receive a response from one reviewer (6FJv) regarding our follow-up question on remaining concerns.

Overall, we appreciate the reviewers’ recognition of several key strengths of our work:

- The **novelty** and **significance** of our method as the "*first zero-shot framework that grounds 3D orientations for multiple objects*" (SEJH, D7en, rW2t).
- The **technical soundness** and elegance of our proposed approach based on the *Langevin-based sampling* and *reward-adaptive time rescaling* (SEJH, D7en, 6FJV, rW2t).
- The **strong empirical results** on our new benchmark (ORIBENCH) and the contribution of the benchmark itself to the community (SEJH, D7en).
- The **high quality of our presentation** and the coherence of our method's logic (SEJH, 6FJV, rW2t).

To address the reviewers’ initial concerns, we also provided additional experimental results during the rebuttal, including:

- Results on additional control inputs (spatial grounding and depth-conditioned generation), demonstrating our method’s versatility.
- Ablations of our Langevin-based sampling versus a naive gradient-ascent method.
- Comparisons with baselines proposed for novel-view synthesis.
- Comparisons with RL-based fine-tuning methods, demonstrating our method’s superiority in both reward alignment and image quality.

For the initial concerns from **6FJv**, although we did not receive a response during the discussion, our rebuttal clarified that the requested justifications or experiments were either (1) included in the main paper (comments 1, 2, 5), or (2) fully addressed with new experiments (comments 3 and 4, covering hyperparameter ablations and comparisons with an RL-based fine-tuning baseline). We therefore believe none of these constitutes a major reason for rejection.

We once again thank the AC and reviewers for their constructive feedback, which has strengthened the manuscript. We will incorporate all suggestions in the revised version to further improve its quality.

---

### Decision · Program_Chairs · 2025-09-17

**Decision:**

Accept (poster)

**Comment:**

The paper presents a method for text-to-image generation with control over the 3D orientation of objects in the generated image. The method leverages an off-the-shelf 3D orientation estimation model, a pre-trained text-to-image model, and achieves test-time orientation control using a guided-sampling approach based on Langevin dynamics, where a sampled latent is chosen to maximize a reward function based on the target orientation.

Reviewers appreciated the technical contributions related to sampling the image latent and the approach to adaptive time rescaling, the general applicability of the method to different models and datasets, the strong results, and the problem setting (zero-shot multi-object orientation control in text-to-image generation). However, they also raised concerns about whether the method can handle variations along the z-axis, the lack of comparisons to more recent baseline methods, the reliance on the pre-trained orientation estimation model, and the requirement for ~50 iterations for image generation.

The authors responded to these concerns in their rebuttal, and three reviewers entered a positive rating. These reviewers noted that their concerns were addressed, or that they would be satisfied with acceptance given the rebuttal results can be included in the paper (which the authors have promised). Although one reviewer remained negative towards the paper, they also noted that the techniques would be valuable for the community. After reading the paper and considering the concerns, I am recommending the paper be accepted, and I encourage the authors to incorporate the results from the rebuttal in the final version.